# MicroRNA governs bistable cell differentiation and lineage segregation via a noncanonical feedback

Chung-Jung Li[1,2] (ID), Ee Shan Liau[1,2] (ID), Yi-Han Lee[2] (ID), Yang-Zhe Huang[2] (ID), Ziyi Liu[3] (ID), Andrew Willems[3] (ID), Victoria Garside[4] (ID), Edwina McGlinn[4] (ID), Jun-An Chen[1,2,5,*] (ID) & Tian Hong[6,7,**] (ID)

## Abstract

**Positive feedback driven by transcriptional regulation has long been considered a key mechanism underlying cell lineage segregation during embryogenesis. Using the developing spinal cord as a paradigm, we found that canonical, transcription-driven feedback cannot explain robust lineage segregation of motor neuron subtypes marked by two cardinal factors, Hoxa5 and Hoxc8. We propose a feedback mechanism involving elementary microRNA–mRNA reaction circuits that differ from known feedback loop-like structures. Strikingly, we show that a wide range of biologically plausible post-transcriptional regulatory parameters are sufficient to generate bistable switches, a hallmark of positive feedback. Through mathematical analysis, we explain intuitively the hidden source of this feedback. Using embryonic stem cell differentiation and mouse genetics, we corroborate that microRNA–mRNA circuits govern tissue boundaries and hysteresis upon motor neuron differentiation with respect to transient morphogen signals. Our findings reveal a previously underappreciated feedback mechanism that may have widespread functions in cell fate decisions and tissue patterning.**

**Keywords** motor neuron differentiation; positive feedback loop; post-transcriptional regulation; single-cell RNA sequencing; tissue boundary formation

**Subject Categories** Development; RNA Biology

**Mol Syst Biol. (2021) 17: e9945**

## Introduction

Systems-level positive feedback serves as a crucial mechanism for cell cycle progression and cell differentiation by generating switch-like behaviors (Xiong & Ferrell, 2003; Novak *et al*, 2007; Yao *et al*, 2008). These switches are often bistable with respect to external signal, and they give rise to cellular memory or irreversibility of cell fate decisions. For example, during embryonic development, bistability arising from feedback loops in gene regulatory networks generates cellular memory with respect to transient differentiation signals such as morphogens (MacArthur *et al*, 2009; Zernicka-Goetz *et al*, 2009). Whereas several important feedback loops that are responsible for irreversible cell cycle progression involve interactions among proteins (Xiong & Ferrell, 2003; Novak *et al*, 2007; Yao *et al*, 2008), the switches in most currently known synthetic and developmental systems are governed by feedback loops mediated by transcriptional activation or inhibition (Gardner *et al*, 2000; Höfer *et al*, 2002; MacArthur *et al*, 2009; Zernicka-Goetz *et al*, 2009; Balaskas *et al*, 2012; Tyson & Novak, 2020).

One critical developmental process that requires robust cell fate decisions is boundary formation between adjacent tissues that ultimately perform distinct biological functions. Differentiating cells near the tissue boundary use positive feedback to make robust cell fate decisions in the presence of competing and/or fluctuating positional signals. The feedback mechanisms often involve a pair of mutually inhibiting transcription factors, which form a double-negative feedback loop (one type of positive feedback loop) (Edgar *et al*, 1989; Cotterell & Sharpe, 2010; Jaeger, 2011; Balaskas *et al*, 2012; Zagorski *et al*, 2017). However, not all known tissue boundary systems have regulatory networks of this type, and it is unclear whether or how robust fate decisions can be made in systems without such canonical transcriptional feedback mechanisms. One such system is the boundary between two types of motor neurons (MNs) along the rostrocaudal (RC; head-to-tail) axis of the spinal cord in bilateral animals. During embryogenesis, antiparallel gradients of retinoic acid (RA) and fibroblast growth factor members (FGFs) along the RC axis determine the expression patterns of Hoxa5 and Hoxc8 paralogs that, in turn, establish the distinct MN identity and synaptic connectivity of rostral limb-innervating lateral motor column (LMC) neurons and caudal LMC neurons at the brachial

---

1  Molecular and Cell Biology, Taiwan International Graduate Program, Academia Sinica and Graduate Institute of Life Science, National Defense Medical Center, Taipei, Taiwan
2  Institute of Molecular Biology, Academia Sinica, Taipei, Taiwan
3  Genome Science and Technology Program, The University of Tennessee, Knoxville, TN, USA
4  EMBL Australia, Australian Regenerative Medicine Institute, Monash University, Clayton, Vic, Australia
5  Neuroscience Program Academia Sinica, Taipei, Taiwan
6  Department of Biochemistry & Cellular and Molecular Biology, The University of Tennessee, Knoxville, TN, USA
7  National Institute for Mathematical and Biological Synthesis, Knoxville, TN, USA
  *Corresponding author. Tel: +886 2 2788 0460; Fax: +886 2 27826085; E-mail: jachen@imb.sinica.edu.tw
  **Corresponding author. Tel: +1 865 974 3089; Fax: +1 949 974 6306; E-mail: hongtian@utk.edu

       *Molecular Systems Biology*   **17**: e9945 | 2021   **1 of 22**

segments (Liu *et al*, 2001; Dasen *et al*, 2003). These neurons form a boundary characterized by mutually exclusive expression of Hoxa5 and Hoxc8 transcription factors, respectively (Dasen *et al*, 2005; Dasen & Jessell, 2009). However, the possible existence of a double-negative feedback loop between Hoxa5 and Hoxc8 was challenged by observations that Hoxa5 does not inhibit Hoxc8 in chicken embryos (Dasen *et al*, 2005; Philippidou & Dasen, 2013; Li *et al*, 2017). Consequently, lack of a known feedback mechanism at the transcriptional level of this system makes it difficult to conceive the mechanism underlying boundary formation between the two MN subtypes. Previous studies have identified two microRNAs, *miR-27* and *miR-196*, that inhibit Hoxa5 and Hoxc8, respectively (Wong *et al*, 2015; Li *et al*, 2017), but their specific roles in cell fate decisions at the MN-type boundary remain unclear.

Here, we used Hox boundary formation in MN subtypes as a paradigm to systematically examine principles of lineage segregation during embryonic development. We found that Hoxa5 and Hoxc8 do not form a feedback loop and their mRNAs exhibit significant cellular overlap, yet they manifested a mutually exclusive boundary at the protein level. To reveal the feedback mechanism underlying lineage segregation, we constructed a series of mathematical models describing elementary biochemical interactions between mRNA and miRNA, which revealed a broad range of biologically plausible parameters that enable bistability through these mRNA–miRNA interactions. Strikingly, these reaction circuits do not exhibit the loop-like characteristics of most known gene regulatory networks. We derived an intuitive conclusion from mathematical analysis, representing a previously underappreciated feedback mechanism arising from a pair of stoichiometric inhibitors with differential degradation rate constants upon formation of multimeric complexes. This feedback loop between miRNA and mRNA molecules does not require transcriptional control or any other canonical feedback mechanism. We estimate that more than $10^4$ distinct instances of this network topology exist in human cells. Using spatiotemporal modeling, embryonic stem cell differentiation, and mouse genetics, we corroborate that (i) *miR-27* is crucial to maintain stable Hoxa5 expression under transient MN differentiation signaling and (ii) *miR-27* and *miR-196* null embryos exhibit MN boundary disruption. Our study uncovered a family of positive feedback loops with widespread molecular interactions that were not previously known to govern bistable switches, and it reveals an unexpected yet potentially general role for miRNA–mRNA interactions in cell differentiation and development.

# Results

## Lineage segregation of Hoxa5 and Hoxc8 in developing spinal motor neurons

To investigate how Hox proteins interpret and respond to gradients of RA and FGF, we first examined distributional dynamics of Hoxa5 and Hoxc8 within spinal MNs along the RC axis of mouse embryos from embryonic days 9.5 to 12.5 (E9.5~E12.5) (Figs 1A and EV1). At E9.5, Hoxa5 was expressed in a subset of cervical segments (prevertebral C4 -> C6), whereas Hoxc8 was absent at this stage. Motor neurons underwent a dynamic boundary formation process from E10.5 to E11.5, with Hoxc8 beginning to be expressed and a small

number of Hoxa5$^{on}$/Hoxc8$^{on}$ double-positive MNs being observed at the Hoxa5-Hoxc8 boundary. At E12.5, Hoxa5$^{on}$/Hoxc8$^{off}$ and Hoxa5$^{off}$/Hoxc8$^{on}$ MNs were sharply segregated into rostrocervical (C4 -> C7) or caudal-cervical (C8 -> T1) segments, and virtually no mixed Hoxa5$^{on}$/Hoxc8$^{on}$ hybrid cells were observed (Fig 1B, $N \geq 3$ embryos). Thus, Hoxa5 and Hoxc8 manifested dynamic yet sharp and robust segregation during spinal MN development in a timely manner, making it a suitable system for interrogating how a gene regulatory network (GRN) can control lineage decisions at a developing tissue boundary (Fig 1C).

## Segregation of Hoxa5- and Hoxc8-expressing spinal motor neurons is not achieved by mutual inhibition

Lineage segregation of cells at several known tissue boundaries, including the cardinal dorsoventral (DV) progenitors along the neural tube, relies on cross-repressive interactions of transcription factors (TFs). Therefore, one of the most conceivable mechanisms potentially underlying the Hoxa5/Hoxc8 lineage decision at the cervical boundary is a canonical feedback loop formed by Hoxa5 and Hoxc8 mutual inhibition (transcriptional cross-repression or T-CR, Fig EV2A) (Cotterell & Sharpe, 2010; Jaeger, 2011; Balaskas *et al*, 2012; Zagorski *et al*, 2017). Nevertheless, the T-CR assumption contradicted the observation from chicken embryos that Hoxc8 unilaterally inhibits Hoxa5 (Dasen *et al*, 2005). To test whether Hoxc8 and Hoxa5 exert mutual or unilateral inhibition in a mammalian context, we generated "Tet-ON" inducible Hoxa5::V5 and Hoxc8::V5 tagged mouse embryonic stem cell (ESC) lines (Fig 2A and B) (Li *et al*, 2017). Under conditions of Hoxa5$^{on}$ MN differentiations (Fig 2A), doxycycline treatment on day 4 of differentiation resulted in efficient induction of exogenous Hoxc8::V5 and concomitant suppression of endogenous murine Hoxa5 expression (Fig 2C, quantification in Fig 2D, $N = 3$ from three independent experiments) (see Materials & Methods for details). Similar to the finding from an avian context (Dasen *et al*, 2005), induction of exogenous Hoxa5::V5 in Hoxc8$^{on}$ differentiated MNs did not repress endogenous Hoxc8 expression (Fig 2E, quantification in Fig 2F, $N \geq 9$ embryoid bodies (EBs) from three independent experiments). These results exclude the possibility of a Hoxa5/Hoxc8 cross-repression circuit (T-CR model).

Next, we built two ordinary differential equation (ODE) models based on the T-CR network and a transcriptional unilateral repression (T-UR) network, respectively (Fig EV2). Transcriptional regulation and cooperativities were described by Hill functions (Appendix Supplementary Methods). We found that the T-CR model exhibited desirable performance in lineage segregation in the presence of noisy morphogen signals described by white noise terms in the ODEs (Fig EV2B–D, Movie EV1), whereas the T-UR model produced fluctuating cell lineages and a blurred tissue boundary under the same condition despite assuming very high cooperativity of gene regulation (Fig EV2E–G and Movie EV2). The poor lineage decision performance of the T-UR model relative to our T-CR model was because the former lacks a positive feedback loop to endow robustness on cell differentiation. In contrast, the mutual inhibition circuit in the T-CR model serves as a canonical form of positive feedback loop to generate bistability and enable hysteresis (Fig EV2H) (Balaskas *et al*, 2012). The difference between the two models was robust with respect to the changes of kinetic rates in the

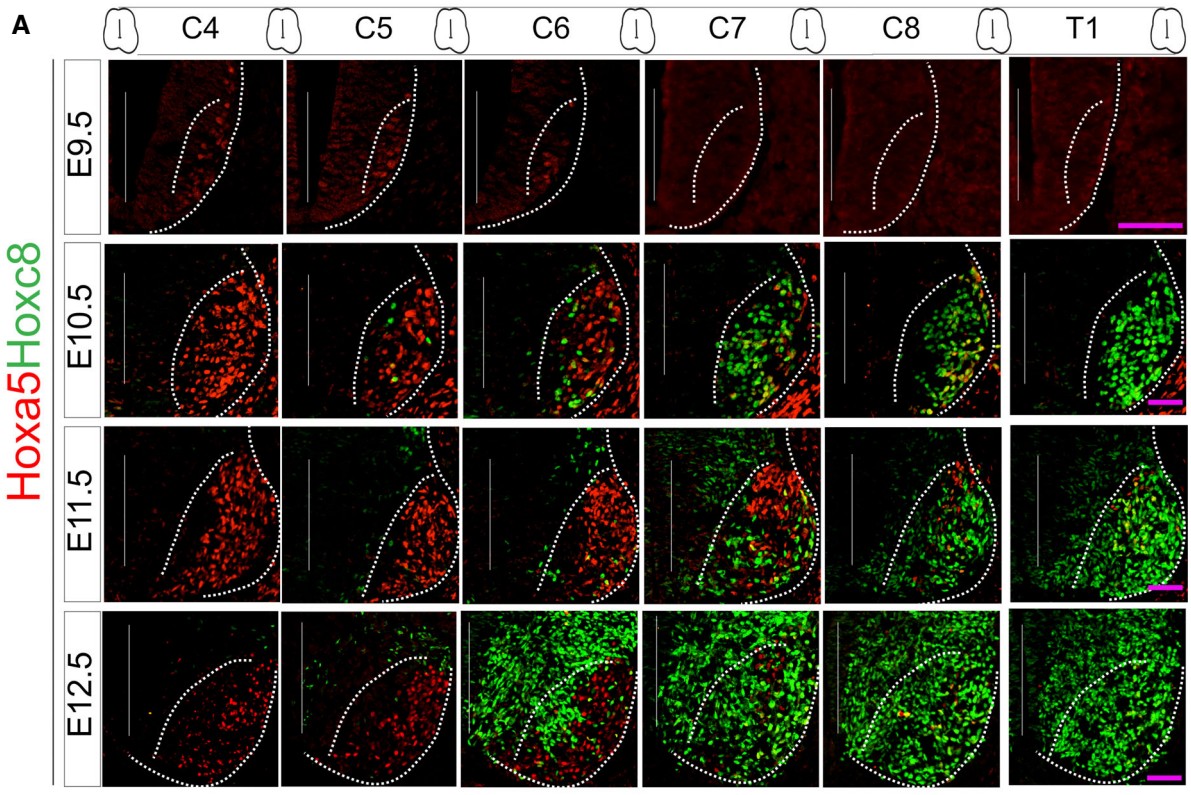

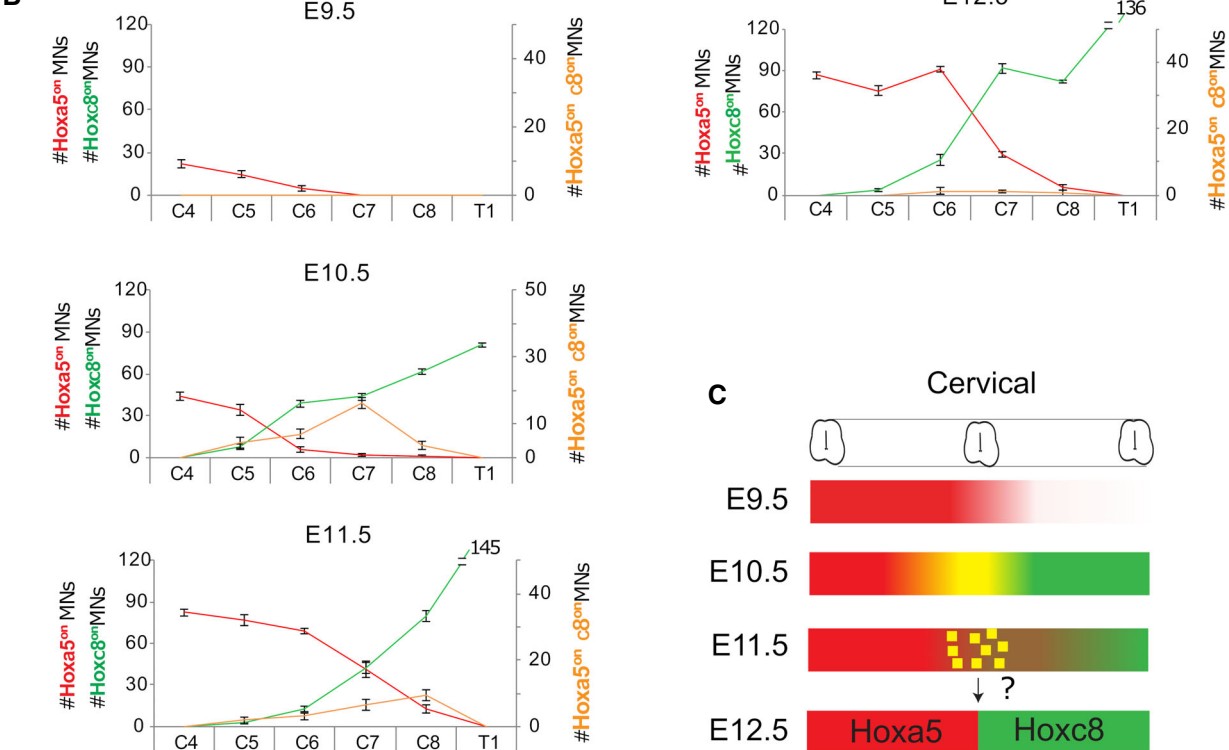

**Figure 1.**

**Figure 1.  Lineage segregation of Hoxa5/Hoxc8-expressing cells in the developing spinal cord.**

A  Expression patterns of Hoxa5 and Hoxc8 in MNs (demarcated by white dashed lines) along the rostrocaudal axis of the spinal cord (cervical C4 to thoracic T1) from mouse embryos from E9.5 to E12.5. Pink scale bars: 12.5 μm for E9.5; 12.5 μm for E10.5; 25 μm for E11.5; and 50 μm for E12.5.

B  Quantification of Hox[on] cells across MN domains. Data represent mean ± SD from $N \geq 3$ embryos.

C  Summary of the intrasegmental expression profiles of Hoxa5 and Hoxc8 in cervical MNs. Yellow squares represent the Hoxa5[on]Hoxc8[on] cells.

models (Fig EV2I). Combined, these *in vitro* and *in silico* analyses indicate that the segregation of Hoxa5[on] and Hoxc8[on] MNs in the face of known inherently noisy morphogen signals (Sosnik *et al*, 2016) may be mediated by a mechanism that differs from the canonical design principle involving mutual inhibition of lineage-determining TFs.

## Hoxa5- and Hoxc8-expressing cells delineate a sharp tissue boundary without segregation of their respective mRNAs

We wondered whether an alternative positive feedback loop mechanism (Delás & Briscoe, 2020) involving other TFs upstream of Hoxa5 and Hoxc8 might function as a potential GRN to explain the

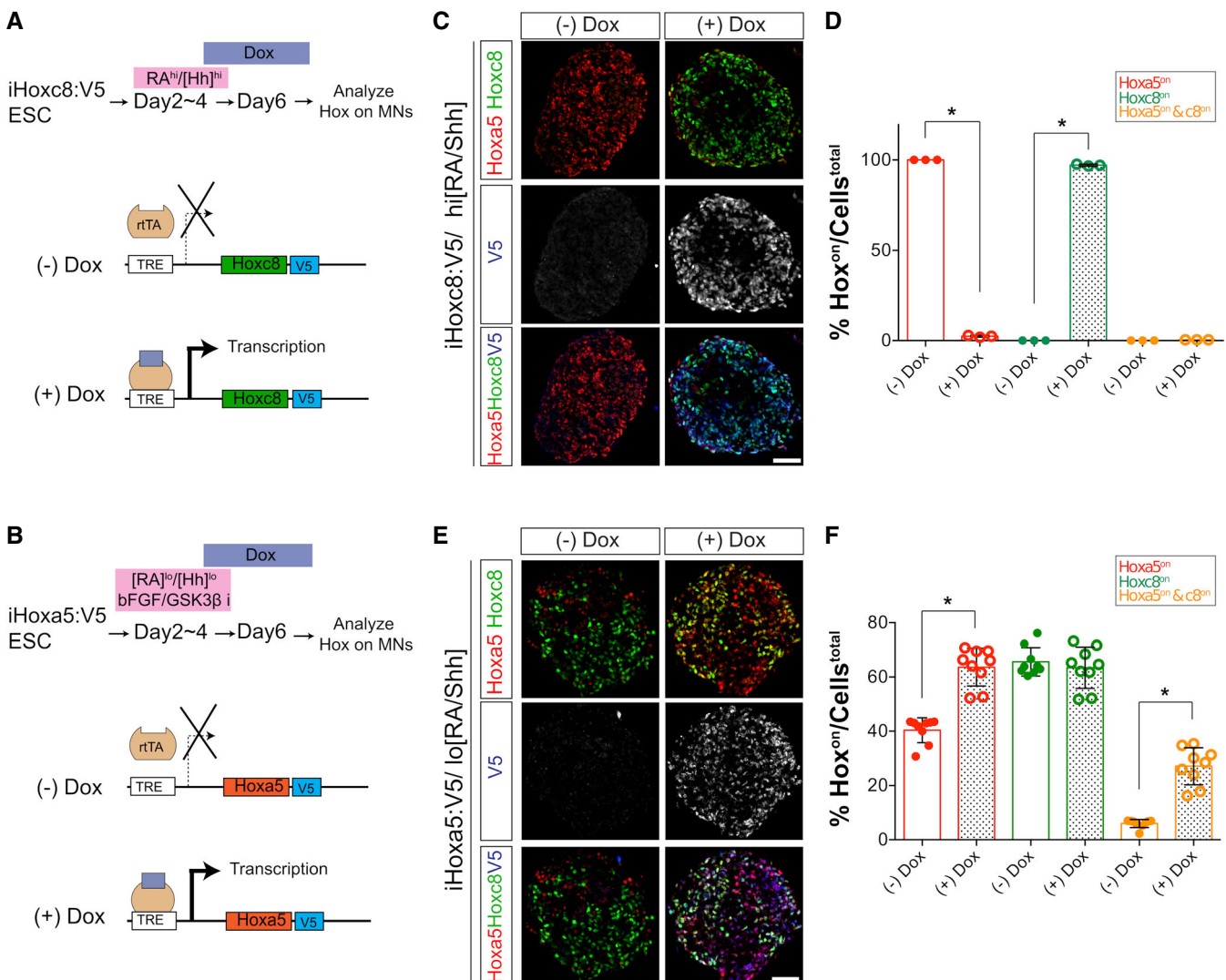

**Figure 2.  Cross-repressive loop is not applicable in the Hoxc8-Hoxa5 lineage segregation in spinal motor neurons.**

A–F  (A, B) Schematic illustrations of the design of inducible "Tet-On" ESC lines expressing Hoxa5 or Hoxc8 under the doxycycline (Dox)-regulated promoter. In the presence of Dox, the reverse tTA (rtTA) activator is recruited to the TRE (tetracycline response element), thereby initiating transcription of the downstream gene. (C, E) Immunostaining reveals expression of Hoxa5 and Hoxc8 upon induction of exogenous Hoxc8:V5 (iHoxc8:V5) or Hoxa5:V5 (iHoxa5:V5), respectively. (D, F) Quantification of data from (C) and (E) (mean ± SD, $N \geq 3$ EBs from three independent experiments, *$P < 0.01$, Student's *t*-tests). Scale bars in (C) and (E) represent 50 μm.

boundary formation of Hoxa5[on] and Hoxc8[on] MNs (Fig EV2J). In this case, a clear all-or-none pattern of *Hoxa5* and *Hoxc8* mRNAs near the boundary at E12.5 would be observed. However, unlike the mutually exclusive pattern of Hoxa5 and Hoxc8 protein expression at the boundary, *in situ* hybridization in the same region revealed largely overlapping *Hoxa5* and *Hoxc8* mRNAs within the cervical spinal cord (Fig 3A). To confirm this observation quantitatively at the individual cell level, we performed single-cell RNA sequencing (scRNA-seq) of cervical MNs collected from Hb9::GFP embryos at E12.5 by fluorescence-activated cell sorting (FACS) (Fig 3B). We clustered single-cell transcriptomes using a graph-based approach, which identified major MN subtypes in the cervical spinal cord (known as motor columns) according to known markers (Appendix Fig S7) (see Materials and Methods for details) (Chen & Chen, 2019). To focus solely on post-mitotic MNs and to simplify our analysis by excluding *Hox* genes poorly or not expressed in other cell types (i.e., MN progenitors, interneurons, or other cell types), we focused on LMC MNs for further characterization given their strong expression of *Hox* genes (Fig 3C) (Dasen *et al*, 2005). Similar to the *in situ* hybridization data, we found that *Hoxa5/ Hoxc8* mRNAs largely overlapped within individual LMC-MN subtypes, whereas Hoxc8-mediated downstream effector genes *Etv4* (*Pea3*) and *Pou3f1* (*Scip*) were disproportionately distributed in *Hoxc8*[+] MNs (Fig 3D) as anticipated (Dasen *et al*, 2005; Catela *et al*, 2016). This result indicates that even though E12.5 cervical MNs have already committed to segregated Hoxa5[on] or Hoxc8[on] lineage fate, *Hoxa5/Hoxc8* mRNAs still exhibited extensive cellular co-expression. To analyze this pattern quantitatively along the RC axis of the cervical spinal cord, we used *Hox4~Hox9* mRNA levels as a proxy to estimate the relative MN positions (Fig 3E) (Philippidou & Dasen, 2013). We found that *Hoxa5* and *Hoxc8* were co-expressed in a large subset of cervical MNs over a broad expanse, with a profound increase in the mid-cervical boundary region defined by *Hoxc4~Hoxa9* expression (Fig 3E). These scRNA-seq results confirm our finding of *Hoxa5/Hoxc8* mRNA co-expression from *in situ* hybridization. Finally, we performed a lineage tracing experiment using the Hoxc8:Cre; ROSA26-loxp-STOP-loxp-tdTomato line (Carroll & Capecchi, 2015) to reveal any cell in which the Hoxc8 genomic locus was, or

had been, active. At E12.5 (Fig 3F), we found that Cre-dependent reporter expression was observed not only in Hoxc8[on] MNs, but also in a significant subset of Hoxa5[on] MNs (Fig 3G, $N = 3$ embryos). This lineage tracing experiment further validated the expression of *Hoxc8* mRNA in Hoxa5[on] MNs of the spinal cord, whereas immunostaining revealed no co-expression of Hoxa5/c8 protein (Fig 3G, $N = 3$ embryos). Together, these results demonstrate that Hoxa5 and Hoxc8 proteins are remarkably segregated despite significant overlap of their respective mRNAs. The data imply a key role for post-transcriptional regulation in the lineage decision process, and they argue against the GRNs in the T-CR and T-UR models, both of which assume segregation of *Hoxa5* and *Hoxc8* mRNAs for tissue boundary formation (Fig 3H).

### Neither a feed-forward loop nor transcription-mediated feedback involving known regulatory miRNAs can explain the Hoxa5-Hoxc8 lineage decision at the tissue boundary

Given that broad importance of miRNAs within Hox GRNs across species (Chen & Chen, 2019) and, more specifically, that previous mouse *in vivo* studies have illustrated that correct spatiotemporal expressions of Hoxa5 and Hoxc8 require *miR-27* and *miR-196*, respectively (Wong *et al*, 2015; Li *et al*, 2017), we investigated whether incorporating these post-transcriptional regulators into our T-UR model could improve its lineage segregation performance and resolve its inconsistency with observed mRNA distributions. We incorporated reported miRNA-Hox GRNs into a new model (i.e., the transcriptional unilateral repression with miRNA regulation, or Tmi-UR, model, Fig EV3A). This new model describes miRNA-mediated mRNA degradation and inhibition, with mass action kinetics similar to previously established approaches (Lu *et al*, 2013; Riba *et al*, 2014) and transcriptional regulation having the same descriptors as for the T-CR and T-UR models. With a representative parameter set, we found that the inclusion of miRNA-based feed-forward loops facilitated the threshold-like behavior of cell differentiation without requiring complete segregation of *Hoxa5* and *Hoxc8* mRNAs (Fig EV3B and C) (Mukherji *et al*, 2011; Osella *et al*, 2011; Li *et al*, 2017). Nonetheless, the Tmi-UR model did not generate positive feedback and, consequently, it did not improve lineage decision or

**Figure 3.  Single-cell RNA-seq and lineage tracing indicate co-existence of *Hoxa5/Hoxc8* mRNAs in MNs along the spinal cord.**

A   Expression patterns of *Hoxa5* and *Hoxc8* mRNAs, as revealed by *in situ* hybridization, aligned with corresponding sections showing Hoxa5 and Hoxc8 immunostainings in MNs (demarcated with pink/white dashed lines) along the cervical spinal cord (cervical C5 to C7) in mouse embryos at E12.5. Pink scale bar, 50 μm. White and pink dash lines demarcate the post-mitotic MN region.

B   Schematic illustration of the strategy to apply single-cell RNA sequencing (scRNA-seq) on FACS-collected homogenous Hb9:GFP[on] MNs.

C   Visualization of E12.5 cervical MNs (right panel) and the subset of LMCs (left panel) using Uniform Manifold Approximation and Projection (UMAP), a 2D nonlinear transformation of high-dimensional data that assigns proximal x-y coordinates to cells (points) with similar expression profiles. Individual cells are colored according to their cluster assignments. Details are provided in the Materials & Methods.

D   Dot-plots showing expression of genes (rows) that distinguish MN subtypes (columns). The height of each colored shape is proportional to the percentage of cells expressing the marker (≥ 1 UMI), and its width reflects the average transcript count within expressing cells.

E   scRNA-seq results reflect that *Hoxa5/Hoxc8* mRNAs largely co-exist along the RC axis of E12.5 spinal cord, particularly at the middle boundary region.

F   Schematic illustration of Hoxc8 lineage tracing. *Hoxc8*-expressing cells and all of their progeny are indelibly marked by tdTomato (RFP) expression in Hoxc8:Cre; ROSA26-loxp-STOP-loxp-tdTomato (lsl-tdTomato) embryos.

G   Analysis of the Hoxc8 lineage (RFP[on] cells) in Hoxc8:Cre;ROSA26-lsl-tdTomato E12.5 spinal cord sections (cervical segments 5–6). RFP is expressed in a broad domain that includes Hoxc8[on] cells and a subset of Hoxa5[on] cells (Hoxa5[on] plus RFP[on], indicated by arrowheads). Pink scale bar, 50 μm. White dash lines demarcate the post-mitotic MN region.

H   Hoxa5[on] and Hoxc8[on] cells adopt an "all-or-none" segregated cell fate despite co-existence of *Hoxa5/Hoxc8* mRNAs, indicating a potential post-transcriptional regulatory mechanism for cell fate segregation.

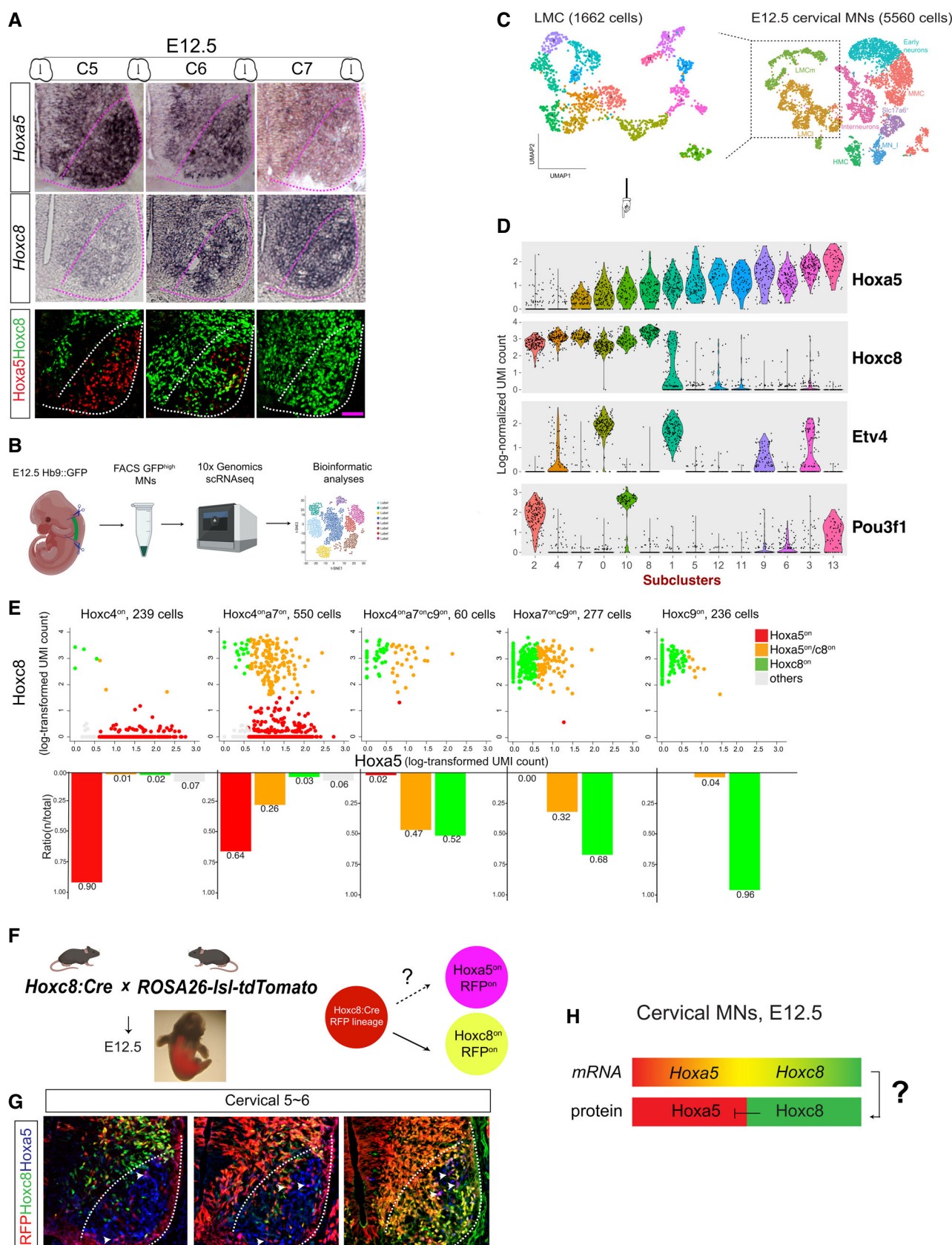

**Figure 3.**

boundary sharpening processes under conditions of fluctuating morphogen signals (Fig EV3C, Movie EV3).

Previous studies on other systems have reported the possibility of a functional TF-miRNA feedback loop whereby the TF transcriptionally inhibits the miRNA that, in turn, represses the TF (Tsang *et al*, 2007; Yoon *et al*, 2011; Lu *et al*, 2013; Palm *et al*, 2013; Tian *et al*, 2013; Zhang *et al*, 2014; Hong *et al*, 2015; Celià-Terrassa *et al*, 2018). Using yet another mathematical model (transcriptional unilateral repression with transcription and miRNA-mediated feedback, or Tmi-FB, model, Fig EV3D), we confirmed that the incorporating two hypothetical transcription-mediated feedback loops between *miR-196* and Hoxc8 and between *miR-27* and Hoxa5, respectively, generated a robust lineage decision at the tissue boundary (Fig EV3E and F, Movie EV4). However, this predicted mechanism requires miRNA expression to be sharply defined. In particular, *miR-196* cannot be expressed in the caudal region in the model (Fig EV3E–H), contradicting the observed distribution of *miR-196*, which expands broadly into the caudal region (Kloosterman *et al*, 2006; Wong *et al*, 2015).

To exclude the possibility that our conclusions on model performance are sensitive to the choice of parameter values, we performed random sampling of parameter values over a wide and biologically plausible range for the T-CR, T-UR, Tmi-UR, and Tmi-FB models (Appendix Table S4), which have 8, 7, 17, or 19 varied parameters, respectively. For each of these models, we randomly chose 10,000 parameter sets and performed simulations using the same procedure. We confirmed that these models either exhibited poor performance in lineage segregation or inconsistency with other experimental observations over a wide range of kinetic rate constants (Fig EV3G–I). Therefore, the lineage decision and boundary sharpening processes of cervical Hoxa5/Hoxc8 MNs are likely governed by a novel post-transcriptional feedback mechanism. In the next section, we describe a new theoretical framework that utilizes the existing gene regulatory network supported by experimental data (Tmi-UR model) to explain the robust segregation of Hoxa5- and Hoxc8-expressing cells (reflecting performance of the T-CR/Tmi-FB model).

## Elementary mRNA and miRNA interactions with simple kinetic requirements can generate bistability

To elucidate the possibility that mRNA and miRNA interactions govern feedback-driven bistable switches, we built a series of mathematical models describing elementary biochemical reactions that involve a pair of interacting mRNA and miRNA molecular species (Fig 4). In these models, a binding event occurs when miRNA(s) bind to an mRNA via sequence complementarity, and they form partially double-stranded complexes (1:$n$ complexes, where $n$ is the number of miRNAs bound to an mRNA). Our models incorporate production of the mRNA and miRNA, as well as their degradation. We assumed that degradation of the complexes involves an elementary step by which either miRNA or mRNA is degraded. This assumption is supported by evidence that both miRNA-mediated mRNA degradation and mRNA-mediated miRNA degradation exhibits multiple turnover (i.e., the miRNA/mRNA can be recycled after the degradation of its binding partner) (Baccarini *et al*, 2011; de la Mata *et al*, 2015). Note that these models still capture possible scenarios in which both mRNA and miRNA are degraded more rapidly (or more slowly) in the complex.

In the first model (mRNA–miRNA with one binding site, or mmi-1, model), an mRNA molecule has one target binding site for a miRNA molecule (Fig 4A). We found that this model cannot give rise to bistability because it is impossible for the system to have more than one steady state for all positive parameter values. For confirmation, we applied the chemical reaction network theory (CRNT) (Feinberg, 2019) to the model and reached the same conclusion (Appendix Supplementary Methods). Next, we considered a model in which the mRNA has two miRNA binding sites (mRNA–miRNA with two binding sites, or mmi-2, model, Fig 4B). Interestingly, we identified a parameter set that supports formation of a bistable switch (Fig 4C, thick curves), an observation consistent with the CRNT and a miRNA model proposed previously (Tian *et al*, 2016), with the latter showing bistability arising from interactions between miRNA and mRNA with multiple binding sites in the absence of transcriptional inhibition. However, the mmi-2 model did not appear to contain any positive feedback loop, an essential element for most known biological switches. Moreover, the range of kinetic rates enabling bistability and their biological plausibility were unclear.

Next, we used a real algebraic geometry method and numerical bifurcation analyses to derive a set of inequalities that describe the ranges of parameters allowing bistability. To examine the intrinsic properties of the regulatory network that enable bistability, we focused on four parameters that describe how fast the mRNA ($a_1$ and $a_2$) and miRNA ($b_1$ and $b_2$) are degraded in the 1:1 and 1:2 mRNA:miRNA complexes, respectively, relative to the degradation rate constants of their single-stranded forms (basal degradation rate constants) (Fig 4B, orange arrows). We found that there is a simple relationship among these rate constants that allows three positive steady states of the system to co-exist (a critical condition for bistability) with certain combinations of the mRNA and miRNA synthesis and basal degradation rate constants (Fig 4C) (a detailed mathematical analysis is included in Appendix Supplementary Methods):

$$\frac{a_1}{b_1} < \frac{a_2}{2b_2}. \tag{1}$$

This analytical result might serve as a simple and general requirement for bistability arising from mRNA–miRNA circuits, but the stabilities of the three steady states, as well as their biological meaning in terms of molecular concentrations, remained unclear. Therefore, we performed numerical experiments to estimate the parameter range allowing for bistability. With parameters $a_1$, $a_2$, $b_1$, and $b_2$ randomly drawn from a uniform distribution over the interval (0.125, 16) (a range estimated from previous experimental work (Eichhorn *et al*, 2014; de la Mata *et al*, 2015)), we identified 33.48% parameter sets that generated bistable switches (Fig 4C, thin curves. Appendix Table S2), and there was no constraint on the ranges of the individual parameters in addition to the chosen interval (Appendix Fig S1). This is consistent with the theoretical prediction based on Eq 1. For most of these bistable switches, increasing miRNA concentrations induced a dramatic decrease in the concentration of free mRNA (> 3 orders of magnitude), and this change was accompanied by a moderate change or no change of the total concentration of mRNA (< 1 order of magnitude) (Fig 4C and Appendix Fig S2). Remarkably, these results suggest that bistable switches may not reflect prominent changes in total mRNA levels

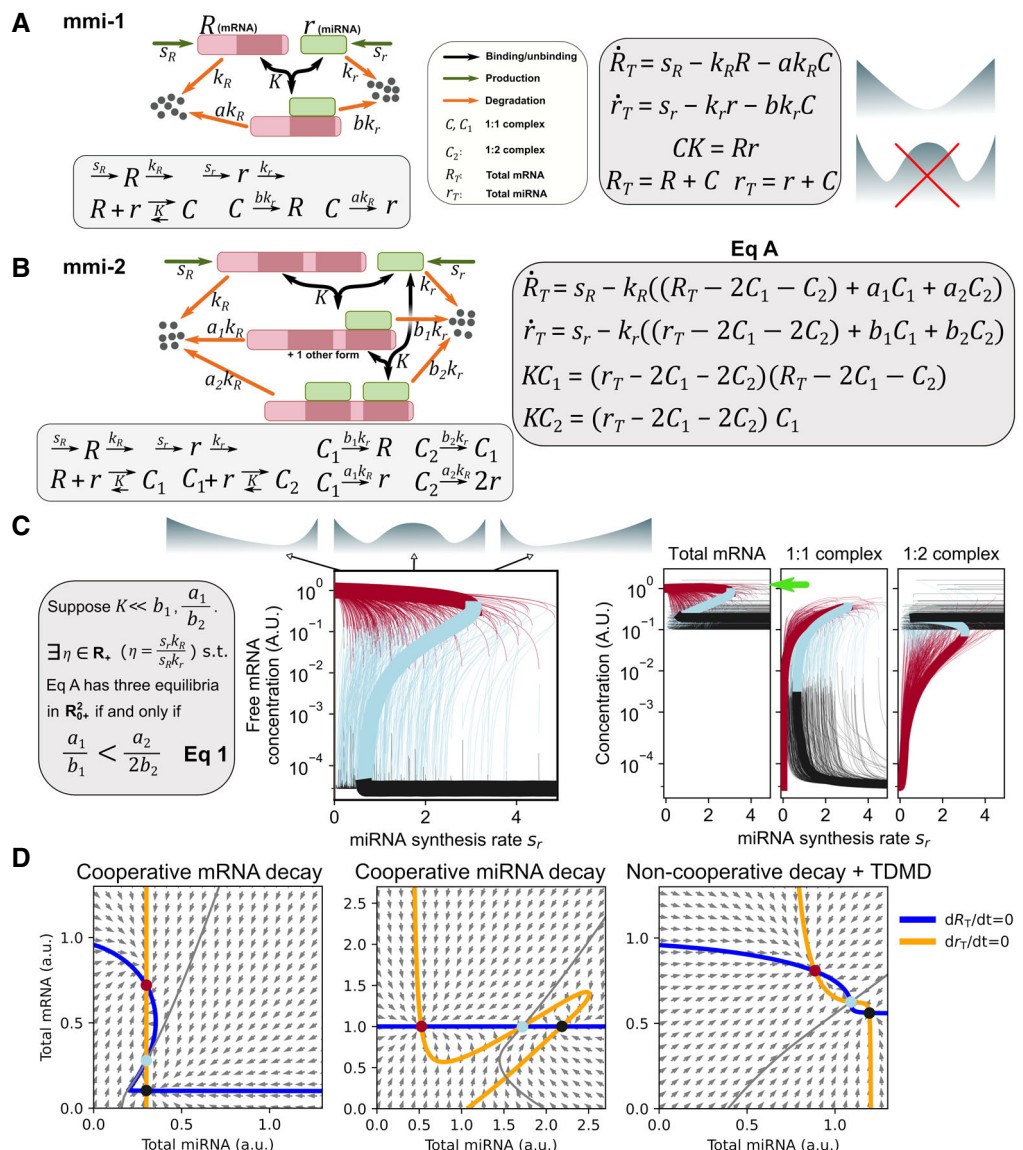

**Figure 4.  Models for cell fate decisions governed by mRNA–miRNA reaction networks.**

A   An mRNA–miRNA reaction network with one miRNA binding site on the target mRNA (mmi-1 model).

B   An mRNA–miRNA reaction network with two miRNA binding sites on the target mRNA (mmi-2 model).

C   Gray box indicates the condition that allows bistability (with Eq A depicted in panel B). We generated 10,000 parameter sets, with values for $a_1$, $a_2$, $b_1$, and $b_2$ randomly drawn from a uniform distribution over the interval (0.125, 16). All other parameters, except the control parameter for bifurcation analysis and the scaled dissociation rate constant ($Kk_R^0/s_R = 10^{-5}$), were set to 1. Bifurcation diagrams demonstrate the steady states calculated for 3,348 bistable systems out of 10,000 parameter sets. Red curve: stable steady state with high amounts of free mRNA. Black curve: stable steady state with low amounts of free mRNA. Blue curve: unstable steady state. One representative system is highlighted with thick curves, and other systems are represented by thin curves. Light green arrow denotes parameter sets in which mRNA degradation is not enhanced by complex formation. In these systems, a steady state is established in the presence of high amounts of miRNA but with total mRNA levels not < 1.

D   Phase plane analysis with three representative parameter sets. Left: cooperative mRNA degradation ($a_1 = b_1 = b_2 = 1$, $a_2 = 10$). Middle: cooperative miRNA degradation ($a_1 = a_2 = b_1 = 1$, $b_2 = 0.1$). Right: target-mediated miRNA degradation (TDMD) ($a_1 = b_2 = 1$, $a_2 = b_1 = 1.8$). All other parameters are equal to 1, except $Kk_R^0/s_R = 10^{-5}$. Dark red and black: stable steady states. Light blue: unstable steady state. Blue curve: nullcline of ODE for $R_T$. Orange curve: nullcline of ODE for $r_T$ (Eq A in Fig 4B). Gray arrows: representative directions in the vector field. Gray curve: separatrix. To obtain the nullclines vector field, coordinates in the $R_T$, $r_T$ space were used as the initial conditions for ODEs in Eq A. The corresponding values of $C_1$ and $C_2$ were obtained by solving the algebraic equations in Eq A with the initial conditions. Resulting values for the right-hand side of the ODEs were used to determine the positions of the nullclines.

Data information: In (A) and (B), light gray boxes show chemical reactions and rate constants, whereas dark gray boxes show the differential-algebraic equations (DAEs) that describe the dynamics of all molecular species.

and that bistability does not require enhanced degradation of both mRNA and miRNA in the complexes. Furthermore, bistability does not seem to require that the degradation rate constants of mRNA and miRNA are in the same range (Appendix Fig S2). Finally, we found that including an additional target (competitor) mRNA in the model did not abolish bistability over a wide range of parameter values (Appendix Fig S5). Nonetheless, Eq 1 and our computational results predict that significantly different mRNA-to-miRNA ratios of degradation rate constants in the 1:1 and 1:2 complexes give rise to bistable switches.

Is this relationship between the two ratios of mRNA and miRNA degradation rate constants (Eq 1, Fig 4C) biologically plausible? We addressed this question by further considering a series of previously reported observations which describe two scenarios of mRNA–miRNA regulations. First, based on multiple turnover of miRNA during target degradation (Baccarini *et al*, 2011), we assume that miRNA degradation is independent of mRNA degradation and that the miRNA degradation rate remains the same in the complexes ($b_1 = b_2$). Grimson *et al* found that target degradation rates arising from two miRNA binding sites are equal to or stronger than the sum of the degradation rates arising from both binding sites individually (Grimson *et al*, 2007) (i.e., $1 < 2a_1 \leq a_2$). The scenario $1 < 2a_1 \leq a_2$ is defined as cooperative miRNA-mediated mRNA degradation. In the second scenario, a previous report indicated that the degradation rates of mRNAs containing multiple miRNA binding sites were significantly increased (2- to 4-fold) with increasing miRNA concentration ($1 < 2a_1 \leq a_2$), whereas this increase was not observed for mRNAs containing single binding sites for the same miRNAs (de la Mata *et al*, 2015). Consequently, miRNA-mediated target degradation is cooperative ($2a_1 < a_2$) or additive ($2a_1 < a_2$) when multiple miRNA molecules bind to the same mRNA molecule, consistent with other studies (Grimson *et al*, 2007). Interestingly, in this scenario, the miRNA degradation rates were reduced with increasing concentration of miRNA ($b_1 > b_2$) (de la Mata *et al*, 2015; Ghini *et al*, 2018). Thus, the effectiveness of target-induced miRNA degradation (TDMD) in mammalian cells is sensitive to the mRNA: miRNA concentration ratio. Therefore, the reverse ratios between the mRNA and miRNA degradation rate constants in different forms of complexes appear to be biologically plausible. Accordingly, based on these previous studies, the simple kinetic relationship we describe in Eq 1 reflects a wide range of realistic parameters. Note that this relationship encompasses other plausible kinetics that allow bistability (e.g., $1 > b_1 > b_2$ or reduced miRNA degradation through multimeric complex formation).

We illustrate the sources of bistability with phase planes for three representative parameter sets corresponding to cooperative mRNA degradation, cooperative miRNA degradation, and TDMD, respectively (Fig 4D). In the first scenario (Fig 4D left), cooperative mRNA degradation from binding with multiple sites gave rise to a Z-shaped response curve (nullcline) governed by the ODE for the total mRNA (Fig 4B, Eq A). In the second scenario (Fig 4D middle), a nonlinear steady-state miRNA level emerged in response to the change in mRNA level. In the third scenario (Fig 4D, right), sigmoidal nullclines for ODEs of both total mRNA and total miRNA were generated from the triggered degradation of both molecules. In each scenario, nonlinear relationships between the two variables gave rise to two stable steady states (nodes, Fig 4D, red and black) and one unstable steady state (saddle point, Fig 4D, cyan). Changes in parameter

values result in coalescence and disappearance of a saddle point and a node, which underlie the saddle-node bifurcations observed for the bistable switches (Fig 4C, both ends of blue curves). Notably, the sources of the nonlinearity in our model differ from several well-known mechanisms crucial for bistability, including cooperativity in binding affinities (Hill functions) (Balaskas *et al*, 2012), saturable enzymes for covalent modifications (zero-order ultrasensitivity) (Goldbeter & Koshland, 1981), and shared enzymes for multiple reactions (Markevich *et al*, 2004). Our model involves the law of mass action for elementary reactions, but not Michaelis–Menten kinetics or any phenomenological nonlinear function.

Finally, we built a model for an mRNA with three miRNA binding sites (mRNA–miRNA with three binding sites, or mmi-3, model, Appendix Fig S3). Through computational analysis with parameter values randomly drawn from a uniform distribution, we found that the fraction of kinetic rate constants that gave rise to bistable switches (50.81%) was even greater than for the mmi-2 model (Appendix Figs S3 and S4). This result indicated that the increased complexity of the reaction network further reduces the requirement for specific rate constants to generate bistability.

Together, the mmi-2 and mmi-3 models demonstrate that a simple and biologically plausible relationship among kinetic rates involved in mRNA–miRNA interactions enables formation of bistable switches. However, the interaction networks (Fig 4B and Appendix Fig S3) do not appear to contain any positive feedback loop, i.e., the network motif considered essential for biological switches and boundary formation (Xiong & Ferrell, 2003; Novak *et al*, 2007; Goldbeter, 2018). This observation prompted us to investigate how to interpret this phenomenon in a context that encompasses a feedback loop and how such a theoretical framework could be used to better understand specific biological systems such as the differentiation of MNs.

### An intuitive interpretation of mathematical analysis reveals a large emerging family of noncanonical feedback loops

Since the condition for bistability involves relationships among degradation rate constants, one possible explanation incorporating a feedback mechanism is that mRNA and miRNA are mutually inhibitive by triggering degradation of each other, thereby forming a double-negative feedback loop. However, this explanation does not reflect the requirement of two forms of complexes, nor does it accurately describe the kinetic relationship (Fig 4C, Eq 1) which suggests that either or both RNA molecules can be "protected" by complex formation, reducing degradation rates, without loss of bistability (Fig 4C and D). We sought to find a unifying and accurate way to resolve this discrepancy by converting the chemical reaction networks into simple influence diagrams, which are commonly used to describe gene regulatory networks using intuitive interpretation.

Since stoichiometric inhibitors are ubiquitous in biology (Ferrell & Ha, 2014; Hopkins *et al*, 2017), it is useful to view the two RNA molecules as a pair of stoichiometric inhibitors (sequesters or sponges) that can mutually reduce their free-form concentrations. Here, the single-stranded (free) forms are considered the "active" forms. For example, in the mmi-1 model, if the degradation rate constants of mRNA and miRNA are balanced in the complex (with respect to the degradation rates of their single-stranded forms) ($a_1/b_1 = 1$), then they simply sequester each other (Fig 5A, left). These

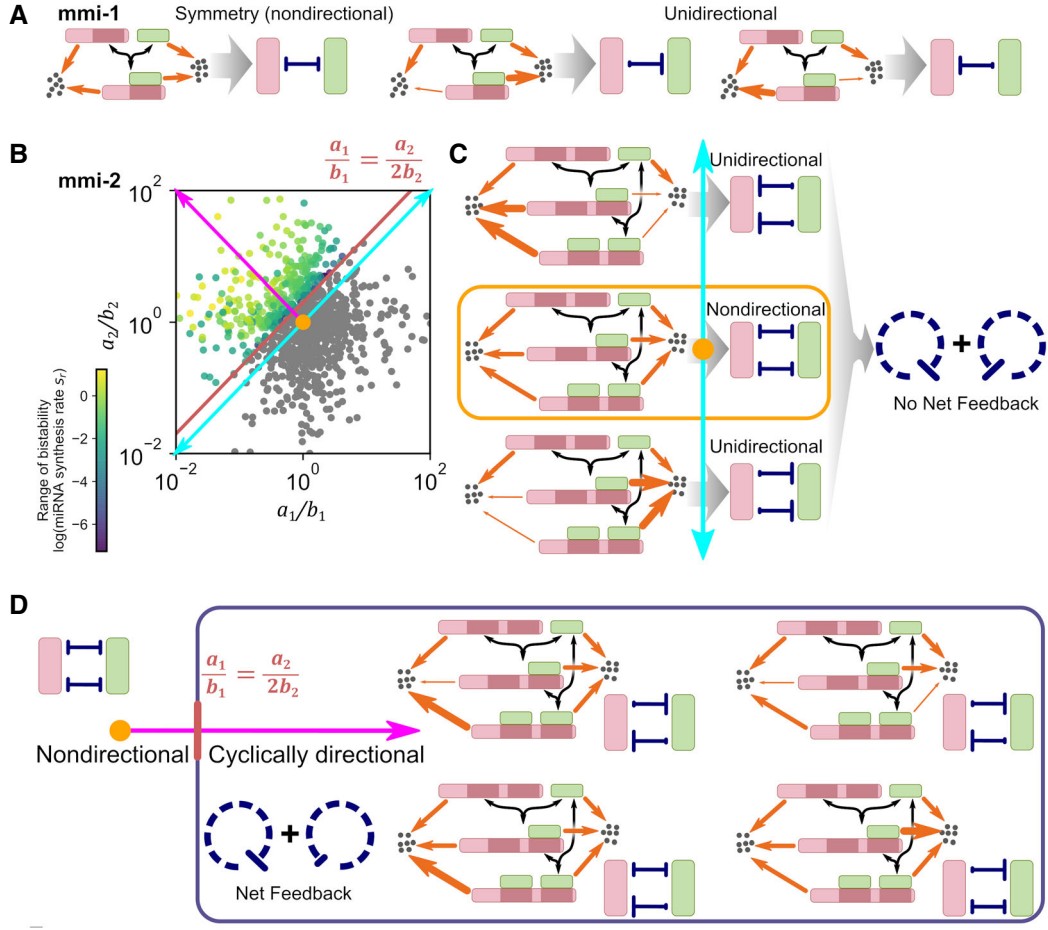

**Figure 5. Theoretical framework for the bistability conditions of mRNA–miRNA reaction networks.**

A  Conversion of the mmi-1 model reaction network into influence diagrams based on two assumptions of kinetics. Left: symmetrical inhibition governed by balanced degradation rate constants in the complex. Right: asymmetrical and unidirectional inhibition governed by unbalanced degradation rate constants between mRNA and miRNA in the complex. Red: mRNA and green: miRNA. Weights of the orange arrows describe the magnitudes of the degradation rate constants.

B  Stability properties of 10,000 randomly generated mmi-2 model systems based on the procedure described in the legend to Fig 4C. Monostable systems (gray dots) and bistable systems (purple-green dots) are shown in the dimensions for $a_1/b_1$ and $a_2/b_2$. The color gradient denotes the range of bistability in terms of the control parameter. Orange dot denotes the condition under which degradation of mRNA and miRNA is balanced in both complexes. Red line is the threshold for bistability predicted by analytical methods. Cyan and magenta arrows serve as visual guides for panels (C and D).

C  Schematic depicting that when both complexes favor the same RNA species (mRNA or miRNA) with unbalanced degradation rate constants, the influences of mRNA and miRNA on each other are asymmetrical and unidirectional. Right diagram shows cancellation of two underlying feedback loops generated by following the inhibition arrows clockwise and counterclockwise.

D  Schematic depicting a spectrum of influences from sponge-like inhibition to feedback-like mutual inhibition. When the influences of mRNA and miRNA on each other become cyclically directional, a feedback loop emerges. Four examples of the kinetic relationships (degradation rate constants) are shown in the box at right.

canonical stoichiometric inhibitions are not feedback loops and they do not generate bistability, as noted in a previous study (Hopkins *et al*, 2017). Furthermore, even if complex formation enhances or reduces degradation of both molecules with the same magnitude ($a_1/b_1$ remains as 1), they are still influenced symmetrically by a single regulatory mechanism. Finally, if the two molecules in the complex exhibit different degradation rate constants relative to their respective free forms ($a_1/b_1 \neq 1$), then the symmetry of the inhibition is broken. Accordingly, the molecule with the lower relative degradation rate constant is a more efficient inhibitor because it is recycled more frequently during degradation events (Fig 5A, right). Consequently, inhibition becomes directional and, for each case of

directional inhibition, complex formation favors either mRNA or miRNA. Therefore, although the regulatory asymmetry is unidirectional, a feedback loop is still absent from the system.

If mRNA and miRNA form two types of complexes (1:1 and 1:2), i.e., the mmi-2 model, the molecular influence of these two types of RNA on each other (underpinned by the reaction network) becomes remarkably diverse and they depend upon the relationships among four relative degradation rate constants (Fig 5B). If inhibition of the mRNA and miRNA is symmetrical ($a_1/b_1 = a_2/b_2 = 1$), the influence of the mRNA and miRNA on each other is nondirectional, albeit governed two complexes. If this symmetry is broken and the unbalanced degradation rate constants of the two complexes favor

the same RNA species (Fig 5B cyan arrow, and C), then the influence is unidirectional. In each of the nondirectional and unidirectional scenarios, we can follow the inhibition influence arrows (Fig 5C, dark blue) clockwise and counterclockwise, resulting in two loops with equal inhibition strengths, which in turn cancel each other out and produce no "net feedback" (Fig 5C, dashed circles). However, if the symmetry of the two inhibitory processes is broken in a cyclically directional manner such that the clockwise loop and the counterclockwise loop do not have equal strengths (Fig 5B magenta arrow and D), then a double-negative feedback loop emerges (Fig 5D). Notably, a defined threshold ($2a_1/b_1 = a_2/b_2$) marks the qualitative transition from symmetrical stoichiometric inhibitions (sequestration) to the feedback loops that enable bistability.

This intuitive explanation informed by influence diagrams captures the essence of the key kinetic relationship that only constrains the ratios of the degradation rate constants between mRNA and miRNA, rather than their absolute values. We found that this explanation of stoichiometric inhibition with broken symmetry also applies to our mmi-3 model, in which each mRNA molecule has three miRNA binding sites, although we were unable to describe the threshold between stoichiometric inhibition and the double-negative feedback loop by means of a simple analytical form. Thus, the formation of bistable switches requires deviation from symmetrical inhibition (Appendix Fig S3B, right, orange dot) and the absence of unidirectional asymmetry of the rate constants (Appendix Fig S3B, left and right, cyan arrows), and both of these requirements are consistent between the mmi-2 and mmi-3 models.

Since the mRNA–miRNA interactions in the mmi-2 and mmi-3 models only represent elementary biochemical reactions, we expect that this network topology is widely deployed in gene regulation. By analyzing predicted miRNA binding sites in human mRNAs (see Methods), we found that our mmi-2 model describes 9,571 mRNA–miRNA pairs and up to 122,885 regulatory networks with distinct binding sites, whereas our mmi-3 model describes 1,250 mRNA–miRNA pairs and up to 93,049 networks (Appendix Table S3). Strikingly, genes that contain two or three targeting sites from the same miRNA mostly belong to cell signaling pathways that are involved in either development or action potential firing processes, which are well known for their irreversible and "all-or-none" characteristics of bistability (Appendix Fig S4C and D). Given the biological relevance of the kinetic relationship (Grimson *et al*, 2007; de la Mata *et al*, 2015; Ghini *et al*, 2018), this analysis implies that the noncanonical feedback mechanism we have derived in this study is applicable to a wide range of gene regulatory networks at the post-transcriptional level.

## miRNA confers hysteresis in the response of Hoxa5 to RA signaling

Next, we assessed whether gene regulation in the MN differentiation system involves networks described by our mmi-2 or mmi-3 models. To do this, we searched for conserved miRNA target sites of Hoxa5 and Hoxc8 using TargetScan (Agarwal *et al*, 2015). We identified three conserved predicted targeting sites for *miR-27* in the *Hoxa5* 3′ untranslated region (UTR), whereas *miR-196* is predicted to have four targeting sites in the *Hoxc8* 3′ UTR (Fig EV4A). Using a

luciferase assay, we tested whether multiple *miR-196* binding sites in the *Hoxc8* 3′ UTR and *miR-27* binding sites in the *Hoxa5* 3′ UTR are critical. Upon overexpression of *miR-196*, we observed ~50% reduction in luciferase activity for the *Hoxc8* 3′ UTR (Fig EV4B). We mutated three of the four binding sites (sites#1–3) individually (Appendix Table S7) and then mutated two and three binding sites sequentially in *Hoxc8* 3′ UTR reporter constructs (Appendix Table S7, Fig EV4B). Interestingly, both site#1 single mutant and sites#2/3 double mutant manifested partially dampened responses to *miR-196* overexpression (Fig EV4B), suggesting a synergy between multiple sites. In addition, combined mutations in sites#1/2/3 triple mutant also led to reduced repression (Fig EV4B).

We used a similar strategy to perform luciferase experiment for *miR-27* binding sites in *Hoxa5* 3′ UTR. Interestingly, we found that mutations in individual sites caused complete de-repression, which was also achieved with double- or triple-site mutation (Fig EV4C). This outcome is consistent with a previous study showing that multiple *miR-27* binding sites in the *Hoxa5* 3′ UTR are required for effective repression (Li *et al*, 2017). Taken together, these results indicate that the multiple mRNA–miRNA interactions satisfy the theoretical requirement (in terms of the network topology) for a bistable switch.

We tested whether MN differentiation exhibits characteristics of a bistable switch. To establish a reliable *in cellulo* model for assaying how Hoxa5 interprets RA signal during ESC differentiation with or without the miRNA/Hox GRN (Figs 6A and EV4D), we treated ESCs with several dosages of RA concentrations from high (1 μM) to medium (250 nM) and low (100 nM), which reliably resulted in a correspondingly proportional high to low number of Hoxa5[on] cells (Fig EV4E and quantification in Fig EV4F). Exposure to [RA][high] for 72 h during differentiation induced robust Hoxa5 expression when assayed at 96 h, representing a stage when Hoxa5[on] MNs become post-mitotic. Interestingly, [RA][high] exposure for only 48 h at the progenitor stage also elicited persistent Hoxa5 expression. Moreover, expression of Hoxa5 was not compromised by switching treatment from [RA][high] to [RA][low] after 48 h, nor by blocking RA signaling at 48 h by addition of the pan RA inhibitor (RARi: AGN 193109, Tocris) to prevent endogenous RA production from EBs, and this cellular memory was recapitulated in the [RA][low] condition (Fig 6B and C, quantification in Fig 6D, $N \geq 3$ EBs from three independent experiments). This experiment demonstrated that expression of Hoxa5 during ESC differentiation manifests hysteresis to transient RA signal, which mimics fluctuating morphogen signal during embryonic development (Sosnik *et al*, 2016).

Subsequently, we examined whether *miR-27* is important to establish the bistable switch of the RA-induced Hoxa5 response. As *miR-27a/b*[−/−] ESCs exhibited poor MN differentiation efficiency (Li *et al*, 2017), we used *iMir27* sponge ESCs (*iMir27SP*) instead to decoy *miR-27* after the progenitor stage (Fig 6E). Both control *iScrmSP* and *iMir27* sponge cells exhibited similar proportions of Hoxa5[on] cells upon [RA][high] treatment (Fig 6F). However, in conditions where [RA][high] was switched to [RA][low] after 48 h (days 0–2), the presence of *iMir27* sponge induced a significant decrease in Hoxa5[on] cells (Fig 6F and quantification in 6G, $N \geq 3$ EBs from three independent experiments). This finding indicates that MN differentiation hysteresis in the presence of transient RA signal was abolished upon loss of *miR-27*. Interestingly, when treated with a medium level (250 nM) of RA for 96 h, more *iMir27* sponge cells expressed

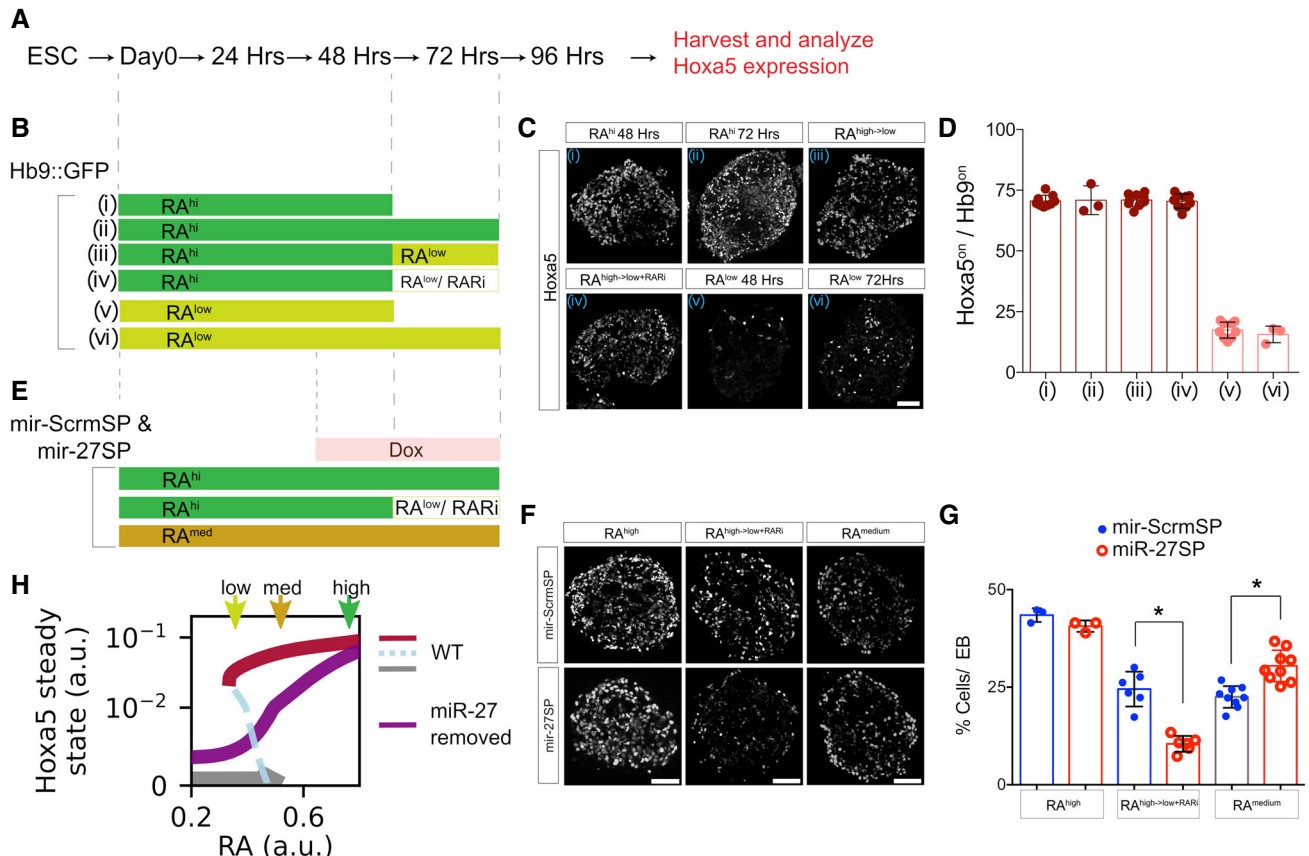

**Figure 6. miRNA confers hysteresis in MN differentiation.**

A, B  Schematic illustrations of the experiments in (C) and (D). Embryoid bodies were incubated with (i) RA$^{hi}$ (1 µM) for 48 h; (ii) RA$^{hi}$ for 72 h; and (iii) RA$^{hi}$ for 48 h, then switched to RA$^{low}$ (100 nM) for 24 h and (iv) RA$^{hi}$ for 48 h, and then switched to RA$^{low}$ plus RAR inhibitor (1 µM) for a further 24 h; (v) RA$^{low}$ for 48 h; and (vi) RA$^{low}$ for 72 h.

C, D  Immunostaining (C) of EBs 96 h after ESC differentiation and quantification (D) of Hoxa5$^{on}$ in MNs (Hb9$^{on}$).

E  Schematic illustration of the experiments in (F) and (G). Inducible ESC lines expressing eight repetitive *mir-27b* sponge sequences were inserted into the GFP 3′ UTR. ESCs were differentiated by doxycycline treatment after 48 h of differentiation. A scrambled sequence was inserted as a control.

F, G  Hoxa5 expression upon *miR-27* knockdown, with exposure to RA$^{hi}$ for 72 h or with exposure to RA$^{hi}$ for 48 h, and then switching to RA$^{low}$ plus RAR inhibitor (1 µM) for a further 24 h or with exposure to RA$^{med}$ for 72 h.

H  Bifurcation analysis of a mathematical model based on our mmi-3 model. Steady states of the system in the presence of *miR-27* (red, gray, and blue) or in the absence of *miR-27* (purple) at various concentrations of RA are shown. Solid curve: stable steady state. Dashed curve: unstable steady state (see Appendix Supplementary Methods for parameter values).

Data information: Scale bar in (C) and (F) represents 50 µm. Data in (D) represent mean ± SD, N ≥ 3 EBs from three independent experiments. The difference between each pair of groups with the same color is not significant (*P* > 0.05); the difference between each pair of groups with different colors (dark red versus pink) show significant difference (*P* < 0.01), Student's *t*-tests. Scale bar in (F) represents 50 µm. Data in (G) represent mean ± SD, N ≥ 3 EBs from three independent experiments, *\*P* < 0.01, Student's *t*-tests.

high levels of Hoxa5 compared to *iScrmSP* cells. This result further implies increased sensitivity of Hoxa5 expression to the change in RA concentration when regulation by *miR-27* is lacking. Not only are these experimental observations consistent with our general theoretical analysis of miRNA–mRNA circuits (mmi-3 model), but they were also captured by a specific set of model parameters describing the RA-induced differentiation of ESCs under both unperturbed and perturbed conditions at three concentrations of RA (Fig 6 H). Therefore, our theoretical framework offers a potential explanation for the robustness of cervical spinal cord patterning. Together, these experiments provide consistent support for our hypothesis that *Hox* expression is resistant to morphogen fluctuation and that

the noncanonical mRNA–miRNA feedback we report here might play an essential role in facilitating cell fate decisions of MNs.

### Noncanonical mRNA–miRNA feedback explains mechanisms underlying motor neuron differentiation and pattern formation in the spinal cord

We wondered whether our theoretical framework of mRNA–miRNA circuitry could explain MN differentiation patterns in the developing spinal cord. To test the possibility that *miR-27* and *miR-196* regulate cell fate decisions at the tissue boundary through positive feedback, we built a spatiotemporal model describing the developing MNs

near the boundary of the Hoxa5 and Hoxc8 expression domains (mmi-S model, Fig 7A). In addition to the antiparallel RA and FGF gradients, the mmi-S model incorporates known *Hox* GRNs, as well as the interactions of miRNAs and mRNA through multiple binding sites, which have been verified experimentally (Fig EV4A) (Li *et al*, 2010; Li *et al*, 2017). We assumed that for each pair of miRNA and

mRNA, the two RNA molecules serve as stoichiometric inhibitors of each other in a cyclically directional fashion, as described above and depicted in Fig 5D and Appendix Fig S6. We fitted this model to observed distributions of *Hox* mRNA, proteins, and their regulatory miRNAs (Figs 1 and 3) (Wong *et al*, 2015; Li *et al*, 2017). The ranges of parameters were the same as those used for all other models

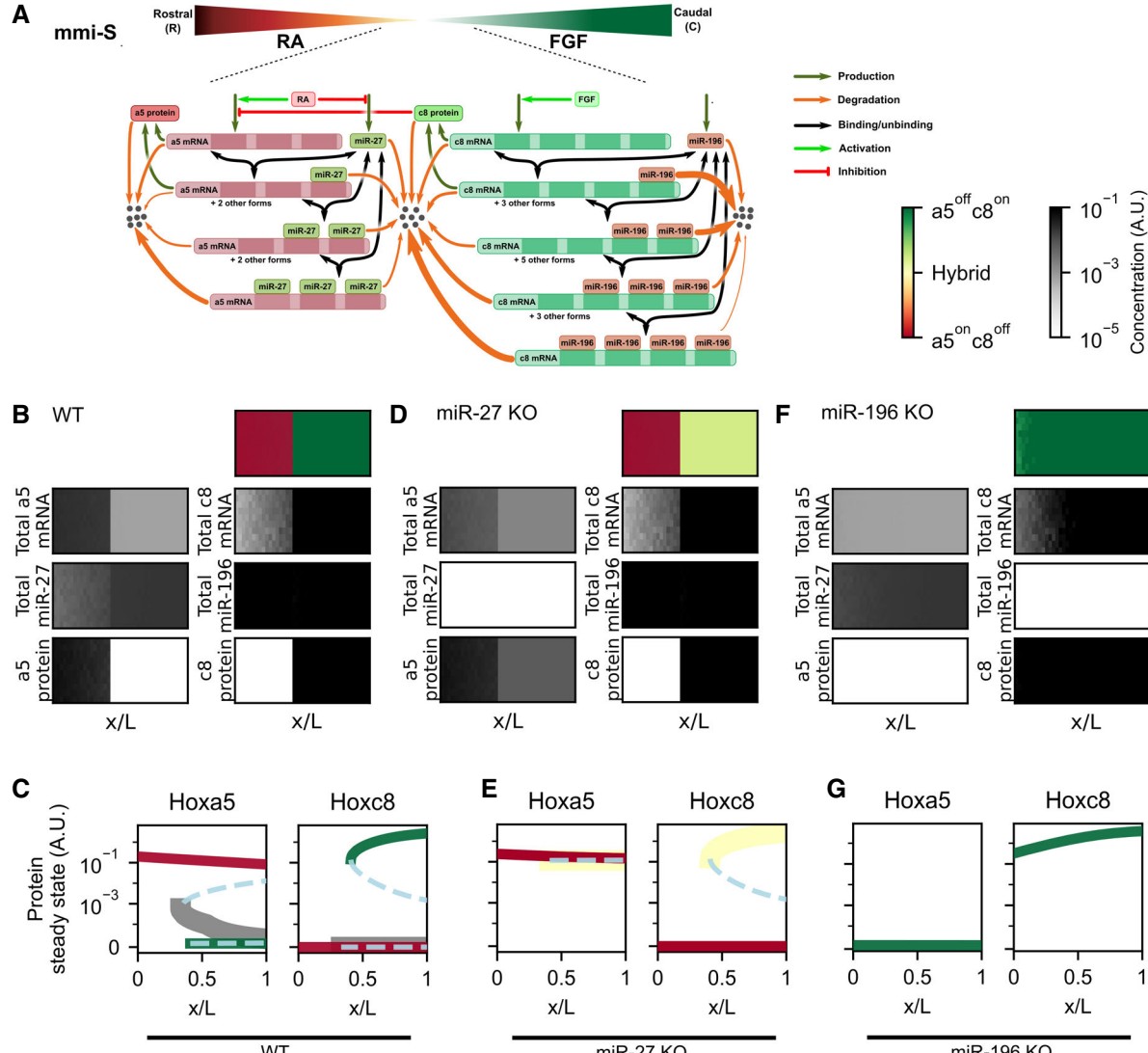

**Figure 7. Spatiotemporal model for rostrocaudal patterning of MN development.**

A   Reaction network for MN development in the RC axis of the spinal cord. Cellular-level reactions including mRNA–miRNA interactions contain known binding sites of *miR-27* and *miR-196* on the 3′ UTRs of the *Hoxa5* and *Hoxc8* mRNAs, respectively. RA and FGF act as signaling molecules to influence transcription. Weights of the orange arrows describe the magnitudes of the degradation rate constants in the mmi-S model.

B   Simulation of the spatiotemporal model. A grid of 10 × 40 cells was used to represent a segment of developing spinal cord where progenitor cells are influenced by competing FGF and RA concentrations. Panel at top right shows the distribution of ratios between Hoxa5 and Hoxc8 protein levels.

C   Bifurcation analysis with position as the control parameter. Solid curves denote stable steady states. Gray: Hoxa5^off Hoxc8^off state; red: Hoxa5^on Hoxc8^off state; and green: Hoxa5^off Hoxc8^on state. The identities of these states were determined by protein levels. Dashed curves denote unstable steady states.

D   Simulation of the spatiotemporal model in the absence of *miR-27*.

E   Bifurcation analysis under the *miR-27* knockout condition. Yellow: hybrid state.

F   Simulation of the spatiotemporal model in the absence of *miR-196*.

G   Bifurcation analysis under the *miR-196* knockout condition.

Data information: Heatmaps in top right corner show the final distributions of denoted molecules in the tissue domain for panels (B), (D), and (F). Panels (C), (E), and (G) show results for low (final) concentrations of RA and FGF (as shown in Fig EV2B).

(Fig EV3I), except for the newly identified feedback mediated by miRNA. Our mmi-S model recapitulated the observations that robust cell fate decisions are reflected in a mutually exclusive expression pattern of Hoxa5 and Hoxc8 at the protein level, but not at the mRNA level (Fig 7B). Interestingly, boundary formation was achieved despite the presence of transient and noisy morphogen signals (Figs EV2B and 7B, Movie EV5). The mmi-S model revealed that this particular MN differentiation system exhibited three types of steady state: Hoxa5$^{on}$Hoxc8$^{off}$ (rostral LMC neuron fate), Hoxa5$^{off}$Hoxc8$^{on}$ (caudal LMC neuron fate), and Hoxa5$^{off}$Hoxc8$^{off}$ (undifferentiated) (Fig 7C). Notably, these three steady states could co-exist when concentrations of both RA and FGF were low, thus forming a tristable system (Fig 7C). This property not only enabled robust fate decisions in the presence of competing morphogen signals, but also provided additional robustness against receding signals. Transient morphogen signals in the developing spinal cord have been observed in both dorsoventral and rostrocaudal patterning due to the pronounced cell proliferation rate and axis elongation during development (Ensini *et al*, 1998; Balaskas *et al*, 2012). In this scenario, the GRN described by the mmi-S model renders progenitors insensitive to signaling fluctuations and confers hysteresis. Overall, the performance of this model in terms of lineage segregation and consistency with experimental data is better than all other models considered in this study (Appendix Fig S6B and C).

To examine the roles of mRNA–miRNA interactions in MN patterning during development, we simulated the mmi-S model in the absence of *miR-27* or *miR-196*. The model predicted that loss of *miR-27* compromised fate decisions and tissue patterning, with Hoxa5 and Hoxc8 presenting overlapped expression at both mRNA and protein levels in the caudal domain (Fig 7D, Movie EV6). This outcome could be attributed to loss of the Hoxa5$^{off}$Hoxc8$^{on}$ steady state, which was replaced by the double-positive Hoxc5$^{on}$Hoxc8$^{on}$ state even under the condition of low morphogen concentrations (Fig 7E). Moreover, our mmi-S model predicted that loss of *miR-196* shifts the Hoxa5-Hoxc8 boundary in the rostral direction (Fig 7F and G, Movie EV7).

Next, we compared our mmi-S model with an alternative model in which each mRNA and miRNA pair do not form a feedback loop according to our theoretical framework (altered mmi-S model, Appendix Fig S6D). Due to noisy RA and FGF signals, cellular identities near the tissue boundary manifested significant temporal fluctuations and were not segregated at the steady state, reflecting the loss of robust cell fate decisions (Appendix Fig S6E). Together, these results indicate that feedback between mRNA and miRNA is essential for the robustness of cell fate decisions near the boundary, as well as for the stability of boundary positioning.

### *miR-27* and *miR-196* govern the sharp and robust Hoxa5-Hoxc8 boundary in the spinal cord

To address whether the predicted function of the miRNA–mRNA circuitry as a bistable switch for Hox proteins explains boundary formation in MNs, we conducted gain-of-function and loss-of-function experiments. First, we developed two "Tet-ON" inducible ESC lines (*imiR-27b* and *imiR-196a*) in which the primary miR-27b or miR-196a sequences were inserted into the 3′ UTR of an inducible GFP construct (Fig EV5A and B). Induced expression of *miR-27a* in

Hoxa5$^{on}$ MNs resulted in efficient suppression of Hoxa5 expression (Fig EV5C, quantification in Fig EV5D, $N \geq 3$ EBs from three independent experiments). Similarly, overexpression of *miR-196a* in Hoxc8$^{on}$ MNs led to a robust reduction of Hoxc8$^{on}$ cells (Fig EV5E, quantification in Fig EV5F, $N \geq 3$ EBs from three independent experiments). In both cases, the generic fate of MNs revealed by Hb9 or Isl1(2) immunostaining was not affected (Fig EV5C and E, quantifications in Fig EV5D and F, $N \geq 3$ EBs from three independent experiments), indicative of specific miRNA–mRNA feedback loops operating separately within the miR-27/Hoxa5 and miR-196/Hoxc8 GRNs.

Next, we investigated whether *miR-27* and *miR-196* are required for the sharp Hoxa5-Hoxc8 boundary in the spinal cord (Fig 8A and B). For *miR-27*, we utilized miR-23–27–24 cluster double-knockout (DKO) mouse embryos (Li *et al*, 2017) to scrutinize the Hoxa5-Hoxc8 boundary segregation process in the spinal cord. A significant number of MNs near the Hoxa5-Hoxc8 boundary in these DKO mouse embryos manifested mixed expression of Hoxa5 and Hoxc8, a cellular phenotype rarely observed in the control embryos at the same stage of development (Fig 8C). Furthermore, we observed intermingled Hoxa5$^{on}$Hoxc8$^{off}$ and Hoxa5$^{off}$Hoxc8$^{on}$ cells in regions where the control embryos showed clear segregation of these two lineages (Fig 8C and quantification in Fig 8D, $N = 3$ embryos), suggesting that the robustness of the cell fate decision is correlated with the robustness of tissue patterning and that miRNAs such as *miR-27* significantly contribute to these phenotypes.

To interrogate the role of *miR-196* in maintaining the precise boundary between Hoxa5 and Hoxc8, we examined Hoxa5 and Hoxc8 protein expression upon loss of all three miR-196 paralogs (196a1, 196a2, and 196b), as they act redundantly to pattern the mid-thoracic region (Fig 8B) (Wong *et al*, 2015). Compared to controls, Hoxc8 protein was expanded profoundly in the rostral Hoxa5$^{on}$ MNs of miR-196 double (miR-196a2/b) and triple (miR-196a1/a2/b) KO embryos at E12.5 (Fig 8E and quantification in Fig 8F, $N = 3$ embryos). Notably, this phenotype is consistent with a report that loss of *miR-196* leads to collective upregulation of numerous *Hox* target genes in the trunk region of mice, which induced impairments in vertebral number and vertebral identity (Wong *et al*, 2015). Furthermore, all of our *in vivo* observations of control embryos or those lacking *miR-27* or *miR-196* are consistent with our predictions from the mmi-S model (Fig 8B–F). Taken together, these findings indicate that miR-27/Hoxa5 and miR-196/Hoxc8 feedback loops are required to maintain a sharp and precise post-mitotic MN boundary in the spinal cord (Fig 8G).

## Discussion

### Role of miRNAs in robust cellular responses

One of the most fascinating questions in biology is how cell fate determination is generally robust in view of fluctuating environmental challenges during embryonic development. Waddington termed this process "canalization", with greater net canalization during development resulting in less phenotypic variation among individuals in a population (Waddington, 1942). Numerous studies have shown that miRNA confers robustness on gene expression through

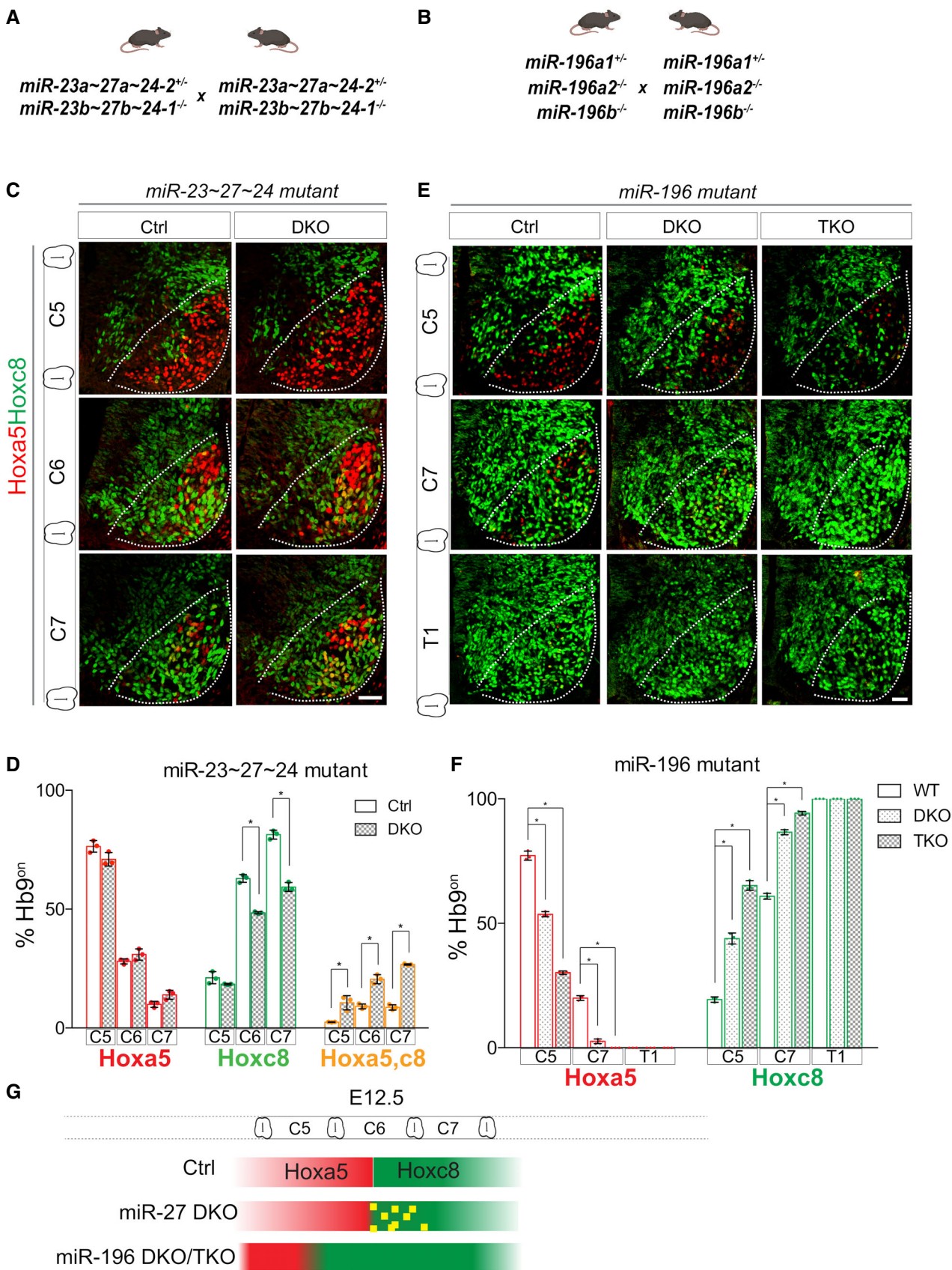

**Figure 8.**

**Figure 8.** *miR-27* and *miR-196* mutants manifest a switched Hox boundary in MNs.

A, B  Schematic illustration of the strategy to generate *miR-27* and *miR-196* knockout embryos.
C–F  (C) Immunostaining at cervical spinal cord sections reveals Hoxa5 expansion into the MNs of cervical C6/7 segments of *miR-23~27~24* double KO (DKO) mice. *miR-23a~27a~24-2^{+/−}; miR-23b~27b~24-1^{−/−}* mice were used as a control (ctrl). (E) Conversely, Hoxc8 is shifted into the MNs of cervical C5 segments of both miR-196a2/b DKO and miR-196a1/a2/b triple knockout (TKO) embryos. Age-matched wild-type embryos were used as controls. Scale bar represents 50 μm. White dash lines demarcate the post-mitotic MN domain (D, F) Quantifications of Hoxa5$^{on}$, Hoxc8$^{on}$, and Hoxa5$^{on}$ Hoxc8$^{on}$ cells in the cervical spinal cord of control and knockout embryos (mean ± SD, N = 3 embryos, *P < 0.01, Student's *t*-tests).
G  Summary of the Hox phenotypes in the miRNA mutants. Yellow squares represent the Hoxa5$^{on}$Hoxc8$^{on}$ cells.

canalization mechanisms, such as feed-forward loops (Li *et al*, 2009; Osella *et al*, 2011; Ebert & Sharp, 2012; Siciliano *et al*, 2013). miRNA can also generate thresholds for gene expression that enhance nonlinearity of the dose response (Mukherji *et al*, 2011). In this study, we found that mRNA–miRNA interactions can generate bistable switches, representing a form of cellular memory, through a feedback loop mechanism that was not previously identified. The simple molecular composition and biologically plausible kinetics of these interactions imply that this feedback loop mechanism could act broadly as a GRN motif to generate cellular memory. Previous studies have shown that combined with TF-mediated miRNA control, miRNA-mediated translational inhibition can be involved in feedback loops (Tsang *et al*, 2007; Lu *et al*, 2013). Our work reveals a distinct feedback mechanism at the post-transcriptional level, which endows robustness on cellular phenotypes in concert with the noise attenuation mechanisms governed by feed-forward loops (Osella *et al*, 2011; Li *et al*, 2017).

### Cooperativity of miRNA-mediated regulations

We have shown that a key component of the mRNA–miRNA feedback mechanism is formation of multiple types of mRNA–miRNA complexes. In addition, this feedback mechanism requires that the complexes have distinct ratios of mRNA and miRNA degradation rates. This kinetic property can be achieved through cooperativity in miRNA-mediated mRNA degradation (de la Mata *et al*, 2015). It also indicates that distinct miRNA molecules can form a feedback circuit with an mRNA if they synergistically reduce the stability of that mRNA. Under this condition, the feedback mechanism does not require multiple binding sites of one type of miRNA on an mRNA, so our theoretical framework can be applied to much broader regulatory circuits involving mRNA–miRNA interactions.

Our proposed kinetic requirement for bistability encompasses, but does not depend on, TDMD, whereby specialized target RNAs selectively bind to miRNAs and induce their decay. Although the detailed mechanism underlying TDMD is not yet entirely clear, structural and biochemical insights indicate that 3′-end miRNA/mRNA complementary sequence matching is a key determinant regulating miRNA activity via 3′-remodeling and/or degradation (Sheu-Gruttadauria *et al*, 2019). In this scenario, *miR-196* and Hoxc8 not only have multiple targeting sites, but also display extended matching 3′-end sequences (Fig EV4A), which might reinforce the reciprocal inhibition between *Hoxc8* and *miR-196*. This topic warrants future verification of potential TDMD of the *Hoxc8*/*miR-196* pair. Experimentally, results from us and others clearly show that (i) Hoxa5 and Hoxc8 have multiple predicted targeting sites for *miR-27* and *miR-196*, respectively (Yekta *et al*, 2004; Li *et al*, 2010; Li *et al*, 2017) and (ii) luciferase assay reveals that expression levels

of Hoxa5 and Hoxc8 are only significantly reduced upon mutation of multiple sites (Yekta *et al*, 2004; Li *et al*, 2017). These results support our modeling that the bistable switch from mRNA–miRNA feedback likely operates via formation of multiple types of mRNA–miRNA complexes. Although we have not established exactly how the dynamic bistable switch of miR-27/Hoxa5 and miR-196/Hoxc8 is generated in terms of rate constants, we have clearly shown that this miRNA–mRNA bistable switch constitutes a cell memory that is robust in the presence of fluctuating environmental signals. It is tantalizing to explore how broadly this miRNA–mRNA bistable switch is exerted in other cell fate decisions during embryonic development.

### Multifaceted functions of stoichiometric inhibition

It was shown previously that whereas the stoichiometric inhibition alone is insufficient to serve as a feedback loop, it often contributes to feedback loop formation (Hopkins *et al*, 2017). Interestingly, simple stoichiometric inhibition can also work as a switch in stochastic systems (Lord *et al*, 2019). Our results show that when kinetics (e.g., the ratio of degradation rates) deviate from a balanced condition in terms of two mutual inhibitors, feedback-like reaction networks can emerge and they can generate bistable switches deterministically. These kinetic variations not only create bistable systems via bifurcation, but they also determine the existence of feedback in a quantitative and continuous fashion. Bistability arising from biochemical reaction networks without explicit feedback loops has been observed based on mathematical models of protein kinase cascades (Markevich *et al*, 2004). Together, these results highlight the importance of kinetics and the structures of chemical reaction networks when analyzing GRNs, and they reveal an important limitation of using signed directed graphs to describe GRNs.

### mRNA–miRNA feedback and the diverse molecular functions of miRNA

The canonical molecular functions of miRNA include mRNA destabilization and translational inhibition (Petersen *et al*, 2006; Bazzini *et al*, 2012; Djuranovic *et al*, 2012). In our models of mRNA–miRNA feedback, a wide range of parameter values enable bistability, such that this feedback mechanism can use both or either of those molecular functions to achieve bistability. This implies that the switch-like property may be a functional trait that has been selected through diverse molecular mechanisms, and it may serve as a unifying performance objective for a broad range of mRNA–miRNA systems, regardless of their molecular functions. Nonetheless, our analysis shows that the feedback mechanism requires strong mRNA–miRNA binding (small *K*) and regulation of the degradation rates in the

complexes, indicating that the canonical functions of miRNA (translational inhibition and mRNA degradation) and TDMD both contribute to how the bistable switch is driven synergistically by the feedback mechanism. Moreover, the relative half-lives for individual miRNAs can vary among cell types (Kingston & Bartel, 2019), implying that the turnover dynamics of mRNA–miRNA during MN differentiation should be considered in our predictive model, which might contribute to the switch between miRNA and target mRNA. Experiments in which the dynamics of miRNA metabolism during spinal MN development are assessed using steady-state metabolic labeling of *mir-27* and *mir-196* would shed light on this topic.

### Tissue-level boundary sharpening

We observed that lineage segregation at the tissue boundary is manifested by proteins rather than mRNAs. Although the detection limits of protein staining may contribute to the absence of Hoxa5 or Hoxc8 signals in some cells, the clearly segregated distributions of these two proteins at E12.5 (Fig EV1), together with upregulation of the downstream factors (Fig 3D), support the lineage commitment of almost all MNs and the formation of a functional boundary. This scenario is in stark contrast to mRNA distributions that do not display any pattern of segregation (Fig 3).

While the performance of our model is similar to that of the known GRNs responsible for robust boundary formation (Goldbeter *et al*, 2007; Cotterell & Sharpe, 2010; Balaskas *et al*, 2012; Zagorski *et al*, 2017), it reveals a previously underappreciated boundary formation mechanism at the post-transcriptional level. Models based on GRNs with feedback loops focus on interpretation of positional signals with autonomous mechanisms for single cells, but sharp boundary formation at the tissue level often involves interactions among cells, including cell migration and induced lineage switching (Cooke *et al*, 2001; Dahmann *et al*, 2011; Addison *et al*, 2018; Tsai *et al*, 2020). Nonetheless, unambiguous cell fate decisions may serve as a foundation for further fine-tuning of tissue boundaries through intercellular interactions. In addition to the feedback mechanism proposed in this study, adopting mRNA–miRNA circuits may allow progenitors to maintain plasticity at the early stage for a diverse choice of subsequent cell fates (Chakraborty *et al*, 2020), and applying the miRNA–mRNA bistable system is advantageous for post-mitotic cells to ensure robustness against fluctuating environments during tissue morphogenesis. We argue that miRNA represents the best arbiter to coherently reconcile these requirements for plasticity and robustness during development. We envisage that future experiments to scrutinize the dynamics of miRNAs and their target interactions at detailed temporal and spatial resolutions by means of single-cell technology will reshape our concept of how cells adopt plastic but robust fates.

## Materials and Methods

### Mouse ESC culture and MN differentiation

*Hb9::GFP, iHoxc8-V5, iHoxc8-V5, imiR-ScrmSP, imiR-27SP, imiR-196a OE,* and *imiR-27b OE* ESCs were cultured and differentiated into spinal MNs as previously described (Yen *et al*, 2018). In some cases, caudal LMC neuron differentiation was achieved by including 100 ng/ml bFGF together with reduced concentrations of RA and

SAG at day 2 of differentiation (Tung *et al*, 2019). All cell lines used in this study are subjected to regular mycoplasma tests.

### Immunocytochemistry

Commercially available primary antibodies and antibodies gifted by J Dasen, H Wichterle, and TM Jessell are described in the table of reagents (Appendix Table S6).

### miRNA *in situ* hybridization

Sections were fixed in 4% paraformaldehyde and acetylated in acetic anhydride/triethanolamine, followed by washes in PBS. Proteinase K treatment was skipped for post-immunostaining. Sections were pre-hybridized in hybridization solution (50% formamide, 5 SSC, 0.5 mg/ml yeast tRNA, 1 X Denhardt's solution) at room temperature, before being hybridized with 3′-DIG or FITC-labeled LNA probes (3 pmol) (LNA miRCURY probe; Exiqon) at 25°C below the predicted Tm value. After post-hybridization washes in 0.2 SSC at 55°C, the *in situ* hybridization signals were detected using the NBT/BCIP (Roche) or Tyramide Signal Amplification System (PerkinElmer) according to the manufacturer's instructions. Slides were mounted in Aqua-Poly/Mount (Polysciences, Inc.) and analyzed by using a Zeiss LSM710 Meta confocal microscope.

### Mouse crosses and *in vivo* studies

Wild-type (WT) C57BL6/J male mice were mated with WT female mice to generate embryos to assess Hox proteins. *miR-23 ~ 27~24* DKO mice were generated by crossing *miR-23a~27a~24-2*[+/−]*;miR-23b~27b~24-1*[−/−] male mice with *miR-23a~27a~24-2*[+/−]*;miR-23b~27b~24-1*[−/−] female mice for experimental analysis. Similarly, the *miR-196* DKO and *miR-196* TKO lines were generated by crossing *miR-196a1*[+/−]*;miR-196a2*[−/−]*;miR-196b*[−/−] male mice with *miR-196a1*[+/−]*; miR-196a2*[−/−]*;miR-196b*[−/−] female mice. Control mice were generated by crossing WT male mice with WT female mice to generate the relative staged embryos for comparison. Mice were mated at the age of 8–12 weeks and the embryo stage was estimated as E0.5 when a copulation plug was observed. Embryos were analyzed between E9.5~E13.5. All of the live animals were kept in an SPF animal facility, approved and overseen by IACUC Academia Sinica.

### Lineage tracing and *in vivo* studies

Hoxc8 lineage tracing experiments were performed by crossing Hoxc8-IRES-Cre mice (a kind gift from M Capecchi's laboratory) with Ai14 (ROSA26-loxp-STOP-loxp-tdTomato) mice (from JAX stock #007908) to generate Hoxc8:Cre;ROSA26-loxp-STOP-loxp-tdTomato (lsl-tdTomato) embryos for experimental analysis. Mice were mated at the age of 8–12 weeks and the embryo stage was estimated as E0.5 when a copulation plug was observed. Embryos were analyzed between at E12.5. All of the live animals were kept in an SPF animal facility, approved and overseen by IACUC Academia Sinica.

### Single-cell sample preparation and RNA-seq

Hb9::GFP (Mnx1::GFP) (JAX stock #005029) transgenic male mice were mated with B6 mice to produce embryos in which all MNs

were labeled with green fluorescence. Mating was confirmed by the presence of a vaginal plug the next morning and defined as embryo stage E0.5. For sample collection, E12.5 mouse embryos were euthanized with $CO_2$ and dissected in Leibovitz's L-15 medium (Gibco, 11415064) to isolate brachial spinal cords (segment C2-T1). Tissue dissociation was carried out by means of enzymatic and mechanical approaches using a Neural Tissue Dissociation Kit (P) (Miltenyi Biotec, 130-092-628) and a gentleMACS Dissociator (Miltenyi Biotec, 130-093-235) according to the manufacturers' instructions. Dissociated cells were resuspended in N2B27/DMEM-F12 and neurobasal medium containing N2 (Life Technologies, 17502048) and B27 (Life Technologies, 17504044), 1% penicillin–streptomycin, 2 mM L-glutamine, 0.2 M β-mercaptoethanol, and 0.5 μM ascorbic acid, supplemented with 1% inactivated fetal bovine serum (FBS), and filtered through a 70-μm strainer (Falcon, 352350). Dissociated cell suspension was subjected to sorting using a BD FACSAria III cell sorter (BD Biosciences, USA) with a 85-μm-diameter nozzle and 45 sheath pressure. $GFP^+$ cells were collected into DMEM plus 1% FBS and immediately processed for cell counting and single-cell isolation. The quality of cells was assayed by measuring live versus dead cells (Thermo Fisher Scientific, L3224) and checked for aggregation using a Countess II Automated Cell Counter (Thermo Fisher Scientific, AMQAX1000).

Single-cell suspension was loaded onto a 10× Genomics Single Cell 3′ Chip (10× Genomics, Pleasanton, CA) and subjected to single-cell isolation, cDNA synthesis, and library construction following the Chromium Single Cell 3′ v3.1 protocol (10x Genomics, PN-1000121). The single-cell library was sequenced using a NextSeq 500/550 platform with pair-ended reads (PE150: Read 1: 28 bp; Read 2: 122 bp).

### Data pre-processing and clustering of cell types

The raw single-cell RNA-seq dataset was processed using the Cell Ranger pipeline (version 3.1.0, 10× Genomics) with standard procedures for demultiplexing, mapping to the mm10 reference via STAR aligner, filtering, barcoding, and UMI counting. The generated expression matrix (cell x gene) was imported into the R platform for downstream analyses.

We performed quality filtering, normalization and scaling, dimensionality reduction, and cell clustering using Seurat package version 2.3.4. Cells fitting the following criteria were retained for further analyses: 1,300–8,000 genes and < 72,200 unique molecular identifiers (UMI), with the upper limits set at the 97[th] percentile, beyond which cells were regarded as outliers. Cells with more than 10% mitochondrial-associated UMI were considered damaged and were removed from the dataset. The dataset was log-normalized and multiplied with a scaling factor of 10,000. Highly variable genes across cells were selected by means of a binned mean-variable plot and subjected to principal component (PC) analysis. To identify significant PCs for clustering analysis, we used both an elbow plot and a jackstraw test (Chung & Storey, 2015). For the former, the elbow lay between the 30[th] and 40[th] PCs, indicating PCs after the first 40 PCs contributed little to overall variation, so we used the first 40 PCs for further analysis. The jackstraw test demonstrated that all 40 initial PCs were significant with a threshold of 0.001. Cells were clustered using the default Louvain algorithm in the FindClusters function with a resolution of 0.18 and visualized on

UMAP. Motor neurons (MNs) and nonmotor neurons were initially distinguished based on expression of generic MN markers such as *Mnx1* and the cholinergic genes *Chat*, *Slc18a3*, and *Slc5a7*. Only clusters expressing MN markers were considered and further analyzed using the same pre-processing procedures as described above, but with a cluster resolution set to 0.28. Motor columns were specifically defined according to their enrichment of known motor column markers. The LMC motor column expressed *Foxp1* and *Aldh1a2* and was further subdivided into $Isl1^+$ (LMCm) and $Lhx1^+$ (LMCl) groups, whereas the MMC motor column expressed *Lhx3*, *Lhx4,* and *Isl1*.

### LMC subclustering

Clusters expressing known LMC markers were defined and clustered as described above, except that the first 30 PCs were used and cluster resolution was set to 0.78. Violin plots were generated to visualize log-normalized UMI counts for motor pool markers such as *Etv4* and *Pou3f1* in individual cells.

### Quantification of the Hoxa5[on]/Hoxc8[on] cell ratio

Our analysis focused on LMC motor neurons. We used the SubsetData function in Seurat to retain annotated LMC clusters for quantification. We binarized expression of *Hoxc4*, *Hoxa5*, *Hoxa7*, *Hoxc8,* and *Hoxc9* based on the frequency distributions of these genes in the dataset. Gene expression was assumed to be bimodal, representing the "on" or "off" populations, respectively. A cut-off to distinguish cells from each population could be set at a local minimum between these two peak populations in the observed single-cell expression distribution. To account for the relatively low expression levels of transcription factors, we used the first local minimum closest to the "off" population. An exception in this regard was made for *Hoxc4*, which was likely to be in excess of bimodal, i.e., having at least three different expression levels of high, low and off. We regarded the high and low $Hoxc4^+$ population as "on" and manually assigned the threshold, while assuming the "low" and "off" peaks were proximal so that identifying a local minimum between the two peaks was nontrivial.

To assign their regional identity, cells were categorized into Hoxc4[on], Hoxc4[on]/Hoxa7[on], Hoxc4[on]/Hoxa7[on]/Hoxc9[on], Hoxa7[on]/Hoxc9[on], and Hoxc9[on] populations based on their binarized expression. Within each of these regions, the Hoxa5[on], Hoxa5[on]/Hoxc8[on], and Hoxc8 populations were calculated as a ratio of the total number of cells. Scatter plots and bar charts were generated using ggplot2 in R (Wickham, 2009).

### Analysis of Hox protein distribution in developing spinal cord

Embryos from various developmental stages (E9.5–E12.5) were obtained from timed matings of WT/*miR-23–27~24* DKO/*miR-196* DKO and *miR-196* TKO/Hoxc8:Cre;ROSA26-loxp-STOP-loxp-tdTomato mice, and detection of a mating plug was counted as embryonic day 0.5 (E0.5). Embryos were dissected out, fixed in 4% paraformaldehyde in PBS for 2 h, and balanced in 30% sucrose after several washes. Fixed embryos were then embedded in OCT compound (Tissue-Tek), frozen in dry ice, and stored at 80°C until use. Spinal sections (20 μm) were made with a CM 1950 cryostat

(Leica) and immediately placed on slides. Based on the total number of sections and the order of each segment from the spinal cord, the precise position of sections along the rostrocaudal axis (from cervical to thoracic spinal cord) could be determined.

### Analysis of Hoxa5 expression in embryoid bodies

ESCs were cultured and differentiated into spinal MNs. Embryoid bodies (EBs) were harvested after applying RA for 96 h and fixing in 4% paraformaldehyde for cryosectioning. Different kinds of Hoxa5 antibodies were used to perform immunocytochemistry, and visualization of Hoxa5 expression was achieved using confocal imaging. The intensity of Hoxa5 protein signal was analyzed using MetaXpress (Multi Wavelength Cell Scoring module, MWCS).

### Generation of inducible "Tet-ON" ESCs

Human HOXC8 and mouse Hoxa5 cDNAs were directionally inserted into pENTR/D-TOPO vector (Life Technology) following manufacturer instructions. Primary miRNA sequence or repetitive miRNA sponge sequence was synthesized and cloned into the 3′ UTR of p2Lox-GFP. Inducible lines were generated by treating the recipient ESCs for 16 h with doxycycline to induce Cre, followed by electroporation of p2Lox-HOXC8:V5/Hoxa5/miRNA OE/miRNA SP plasmids. After G418 selection, individual resistant clones were picked and characterized. After 10–15 days of selection, clones were expanded. Details of primer and miRNA sequences are provided in SI Table S5. Inducible miRNA overexpression and sponge ESCs were cloned into the 3′ UTR of the p2Lox-GFP construct, and the same procedure as described above was followed to generate stable ESC clones.

### Dual-luciferase reporter assay

The WT 3′-UTR sequence containing four putative miR-196 binding sites of the *Hoxc8* gene was amplified from mouse genomic DNA by PCR using 2X PCR Dye Master Mix II (ADPMX02D-100) or Phusion High-Fidelity DNA Polymerase (F-530L; Thermo Fisher Scientific). The purified PCR products were then inserted into the psiCHECK™-2 vector (C8021; Promega) at *XhoI* and *NotI* restriction sites by using T4 DNA ligase (M0202S; NEB). The mutated versions (Mut) of the *Hoxc8* 3′-UTR were cloned by designing primers that resulted in the complementary sequence of the predicted binding site mismatching the *miR-196* seed sequence. Each fragment possessing a mutated binding site was initially amplified by PCR, followed by overlapping extension PCR, and the two or three mutated binding sites within the *Hoxc8* 3′ UTR were then cloned into the psiCHECK™-2 vector described above. The primers used for the WT and Mut *Hoxc8* 3′ UTR reporter are listed in Appendix Tables S7 and S8.

For *Hoxc8*, HeLa cells were plated at a density of $1 \times 10^4$ per well (96-well plate), expanded for 16–20 h, and co-transfected with a mixture of 50 ng of WT or Mut reporter and 350 ng of either control *mir-Scramble* or *mir-196a* plasmids using 0.8 μl of PLUS Reagent and 0.4 μl of Lipofectamine LTX Reagent (A12621; Invitrogen). After 24 h, cells were lysed and processed for luciferase assay using the Dual-Luciferase Reporter Assay System (E1910; Promega) according to the manufacturer's instructions. We measured luciferase activity using a 20/20n luminometer (Turner Biosystems). To normalize transfection efficiency, luciferase activity was calculated as the ratio

of firefly to Renilla luciferase activity, and the relative luciferase activity was further expressed as the ratio of measured luciferase activity to the control (Chen *et al*, 2011; Tung *et al*, 2015).

For Hoxa5, conditions were the same as Hoxc8 experiments, except that the experiment was performed in ES cells with condition: 50 ng of WT or Mut reporter and 350 ng of either control *mir-Scramble* or *mir-196a* plasmids using 0.8 μl of PLUS Reagent and 0.4 μl of Lipofectamine LTX Reagent (A12621; Invitrogen) (Li *et al*, 2017).

### Statistical analyses and graphical representations

All statistical analyses were conducted using GraphPad Prism 6 (GraphPad Software). Values are presented as mean ± SD, as indicated. Student's *t*-tests were used for comparisons between experimental samples and controls. Statistical significance was defined as *$P < 0.01$ by Student's *t*-test unless otherwise indicated.

### Mathematical models for gene regulation and reaction networks

Details of the gene regulatory network models are included in Appendix Supplementary Methods. For mRNA–miRNA reaction network models, we described production, degradation, and binding of mRNA ($R$) and miRNA ($r$) molecules with ordinary differential equations (ODEs) using mass action kinetics. Our mmi-1, mmi-2, and mmi-3 models describe mRNA ($R$) containing one, two, or three binding sites for miRNA ($r$), respectively. Since binding and unbinding processes are much faster than production and degradation, we adopted a total quasi-steady-state assumption (Borghans *et al*, 1996) and reduced the ODEs with 3–5 state variables to only two ODEs describing slow processes and 1–3 algebraic equations describing fast processes. We analytically showed that the mmi-1 model cannot be bistable for all parameters. We analyzed the mmi-2 model using an algebraic-geometric approach to obtain a relationship among parameters that allow three steady states, a necessary condition for bistability (Eq 1). We confirmed the relationship using numerical methods, including simulations and numerical bifurcation analysis. For the mmi-3 model, we used numerical approaches to find the range of parameters that allow bistability. Details of the mathematical analyses are included in the Appendix Supplementary Methods.

### Chemical reaction network theory

We applied chemical reaction network theory (CRNT) (Feinberg, 2019) to assess the ability of the mmi-1, mmi-2, and mmi-3 models to generate bistability. The mmi-1 model is a deficiency zero system, so it is monostable for each combination of positive parameter values. The mmi-2 model is a deficiency one system, so it is bistable for certain combinations of positive parameter values. Although CRNT analysis did not generate a definitive conclusion for the mmi-3 model, its subnetwork contains the mmi-2 model structure so it can also generate bistability for some combinations of positive parameter values (Conradi *et al*, 2007).

### Enumerating mRNA–miRNA reaction network motifs

To estimate how frequently the mRNA–miRNA reaction network motif represented by the mmi-2 model can be found in biological

systems, we obtained a dataset on predicted miRNA binding sites in humans from TargetScan (Agarwal *et al*, 2015). To estimate the lower bound of the number of incidences of the motif, we counted the number of mRNA–miRNA pairs in which the target mRNA has two or more conserved binding sites for the cognate miRNA. To estimate the upper bound, we counted the number of miRNA binding site pairs (conserved and nonconserved), each of which share a target mRNA and cognate miRNA.

### Spatiotemporal model for MN development

We built spatiotemporal models for a group of developing MNs near the boundary between the Hoxa5- and Hoxc8-expressing regions. We modeled a grid containing $10 \times 40$ cells, where 40 is the number of cells along the rostrocaudal axis. In the models, diffuse RA and FGF molecules provide positional information along the rostrocaudal axis, which are produced at the rostral and caudal boundaries of the domain, respectively. We assumed uniform degradation rate constants for RA and FGF in the domain, so the cells were influenced by exponential functions of RA and FGF gradient concentrations along the rostrocaudal axis in an antiparallel fashion. For each cell in the mmi-S model, the spatiotemporal model describes the dynamics of Hoxa5 and Hoxc8 (mRNAs and proteins), as well as miR-27 and miR-196, under the influence of both RA and FGF. Key interactions of the GRNs include that (i) RA activates Hoxa5 transcription (Liu *et al*, 2001; Li *et al*, 2017); (ii) FGF activates Hoxc8 transcription (Liu *et al*, 2001); (iii) Hoxc8 inhibits Hoxa5 transcription (Dasen *et al*, 2005; Philippidou & Dasen, 2013); (iv) miR-27 inhibits Hoxa5 translation by binding to two or more sites at the 3′ UTR (Li *et al*, 2017); and 5) miR-196 inhibits Hoxc8 translation by binding to the 3′ UTR (Wong *et al*, 2015). We also assumed that miRNA–mRNA interactions occur at three predicted *miR-27* binding sites on *Hoxa5* mRNA and at four predicted *miR-196* binding sites on *Hoxc8* mRNA (as predicted by TargetScan) (Agarwal *et al*, 2015), although only two of those binding sites in each mRNA–miRNA pair are essential for our conclusions. Influence diagrams of four alternative spatiotemporal models, i.e., T-CR, T-UR, Tmi-UR, and Tmi-FB models, are shown in Figs EV2A and E, and EV3A and D respectively.

We performed numerical bifurcation analyses and simulations on these spatiotemporal models. The parameter values were chosen from randomly sampled sets and adjusted manually (see Appendix Supplementary Methods). Since mathematical analysis of the mmi-1, mmi-2, and mmi-3 models revealed that a simple relationship among kinetic rates is sufficient to obtain switch-like behaviors, we chose parameter values for the mmi-S model such that the Hoxa5$^{on}$ cells and Hoxc8$^{on}$ cells are differentiated by activation of two bistable switches governed by two pairs of mRNA–miRNA feedback loops, respectively. Accordingly, the qualitative behaviors of the model could be reproduced using a wide range of parameter values. Simulated cells were induced to express low levels of Hoxa5 and Hoxc8 before activation of RA and FGF signaling through morphogen production and diffusion. Steady-state distributions of Hoxa5$^{on}$ cells, Hoxc8$^{on}$ cells, and Hoxa5$^{on}$ Hoxc8$^{on}$ cells in terms of both protein and mRNA levels were analyzed and compared experimentally. *miR-27* and *miR-196* knockouts were simulated by turning off their productions in all cells. List of all models and their parameter values is included in Appendix Tables S1, S4 and S5.

## Data availability

The datasets and computer code produced in this study are available in the following databases:

- Modeling computer scripts: GitHub (https://github.com/lfsc507/mmi).
- Single-cell RNA-sequencing data: Gene Expression Omnibus GSE156023 (https://www.ncbi.nlm.nih.gov/geo/query/acc.cgi?acc=GSE156023).

Expanded View for this article is available online.

## Acknowledgements

We thank the Genomic, FACS, and Imaging cores of IMB, Academia Sinica, for considerable technical help. The Hoxc8-IRES-Cre line was a kind gift from Prof. Mario Capecchi from the School of Medicine, University of Utah. The miR-23–27~24 DKO mouse founder strains were generated by the Transgenics Core Facility of IMB. We thank J. Dasen (NYU) for the gift of Hoxa5 antibody and H. Wichterle (Columbia University) for giving us Hb9 antibody and Hb9::GFP ESCs. We also acknowledge Y.-H. Su and S.-J. Chou (ICOB, Academia Sinica) for discussions on our experimental results, members of the JAC laboratory members for proofreading, Mien Chang for mouse strain maintenance, and J O'Brien for further reviewing the manuscript. We thank X. Cheng and D. Handwerk for helpful discussions and suggestions. This work was supported by the HHS|NIH|National Institute of General Medical Sciences (NIGMS) of the National Institutes of Health under Award Number R01GM140462 to T.H. This work was supported by Academia Sinica (CDA-107-L05 & AS-GC-109-03), Ministry of Science and Technology, Taiwan (MOST) (109-2314-B-001-010-MY3), and National Health Research Institutes (NHRI) (NHRI-EX110-10831NI).

## Author contributions

Conceptualization: J-AC and TH; Methodology: C-JL, ESL and TH; Modeling analysis: ZL, AW and TH; Experiments: C-JL, ESL, Y-HL, Y-ZH and J-AC; Resources: VG and EM; Writing – original draft: J-AC and TH; Writing – review & editing: J-AC and TH; Funding acquisition: J-AC and TH; Supervision: J-AC and TH.

## Conflict of interest

The authors declare that they have no conflict of interest.

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
