## [Review Process File · Molecular Systems Biology]

MicroRNA governs bistable cell differentiation and lineage segregation via a noncanonical feedback

Chung-Jung Li, Ee Shan Liao, Yi-Han Lee, Yang-Zhe Huang, Ziyi Liu, Andrew Willems, Victoria Garside, Edwina McGlenn, Jun-An Chen, and Tian Hong

DOI: [10.15252/msb.20209945](https://doi.org/10.15252/msb.20209945)

Corresponding author(s): Tian Hong (hongtian@utk.edu), Jun-An Chen (jachen@imb.sinica.edu.tw)

Review Timeline:

Submission Date:	25th Aug 20
Editorial Decision:	1st Oct 20
Revision Received:	18th Feb 21
Editorial Decision:	14th Mar 21
Revision Received:	21st Mar 21
Accepted:	23rd Mar 21

Editor: Jingyi Hou

Transaction Report:

Thank you for submitting your work to Molecular Systems Biology. We have now heard back from the three reviewers who agreed to evaluate your manuscript. As you will see from the reports below, the reviewers acknowledge the potential interest of the study. They raise however a series of concerns, which we would ask you to address in a major revision.

Since the reviewers' recommendations are rather clear, there is no need to reiterate all the points listed below. Some of the key issues that would need to be addressed are the following:

- Reviewer #1 is concerned about the experimental validation of the proposed mechanism and makes constructive suggestions in this regard. This issue needs to be convincingly addressed.
- Reviewer #2 suggests splitting the study into two separate manuscripts, which we would not recommend. However, efforts should be made to make the current manuscript more concise and accessible to the general readers of Molecular Systems Biology.
- The essence of the model, the underlying assumptions of the model, and the mechanism underlying bistability need to be better explained.
- Attention should be given to placing the findings in the context of existing literature, as commented by Reviewer #3.

All other issues raised by the reviewers need to be satisfactorily addressed as well. As you may already know, our editorial policy allows in principle a single round of major revision and it is therefore essential to provide responses to the reviewers' comments that are as complete as possible.

On a more editorial level, we would ask you to address the following issues.

Reviewer #1:

The establishment of different cell fates and sharp gene expression boundaries is critical for correct animal development. In this manuscript Li et al. utilize in-vivo developmental biology and stem-cell based in-vitro differentiation approaches in combination with computational modelling to achieve a mechanistic understanding of the underlying molecular principles by which Hoxa5-positive cervical and Hoxc8-positive brachial motor neurons arise in the developing spinal cord. Using immunofluorescent stainings for Hoxa5 and Hoxc8 they first study the protein dynamics that underlies the formation of this boundary in the developing mouse spinal cord. They then demonstrate that besides the formation of a sharp boundary at the protein level, the mRNAs of both Hox genes are co-expressed in a broad area of the developing spinal cord. This observation, in combination with previous observations that miRNAs are required for the correct expression of these Hox genes, led them to investigating the importance of post-transcriptional mechanisms in this process. Using computational modelling the authors demonstrate that 1) bi-stability can arise from mRNA-miRNA interactions if an mRNA harbours more than 1 binding site for a miRNA and the degradation rate constants follow a simple relationship and 2) that this bi-stability emerges as a consequence of negative feedback loops appearing due to the symmetries of the different configurations (unbound, bound by 1 miRNA, bound by 2 miRNAs, ...) being broken in opposite directions. They experimentally test if this mechanism can explain the formation of the Hoxa5 - Hoxc8 gene expression boundary by demonstrating that their model can account for the hysteresis in response to retinoic acid and the phenotypes observed upon different genetic manipulations in stem cell-derived motor neurons and the developing mouse embryo.

Overall this is a well-executed study that addresses an interesting topic - how different cell fates are robustly established during embryogenesis despite the presence of inherently noisy patterning systems. The authors combine computational modelling with experimental perturbations to demonstrate an important role of miRNAs during motor neuron subtype specification in the developing spinal cord and propose a conceptually novel mechanism how broadly expressed miRNAs can contribute to the formation of sharp gene expression boundaries. As miRNAs have important functions during all kind of processes ranging from development to tissue homeostasis and disease, I believe these findings will be interesting for a broad audience. While I am not the best person to judge the validity of the math underlying their modelling, the modelling section appears convincing and the authors clearly made an effort to explain their findings in a way that it can be understood by a broad audience. The experimental biology extends previous observations from the authors and other groups how miRNAs contribute to correct motor neuron patterning and subtype specification. The authors furthermore provide a high-quality single cell RNA sequencing dataset from embryonic motor neurons, which will be of broad interest to researchers working on spinal cord development or on the generation of motor neurons from stem cells, e.g. for therapeutic approaches. For these reasons I believe this manuscript will be a valuable addition to the field.

Major comments:

1) The key outcomes of the modelling section are that for the emergence of bistability 1) multiple

miRNA binding sites are required (lines 252-254, Figure 5A) and 2) the inhibitory processes for different miRNA-mRNA configurations are broken in opposite directions (lines 366 - 371, Figure 6D). Although the authors provide novel evidence that miR-27 and miR-196 interact with Hox genes and are required for correct motor neuron subtype specification, they never directly test these predictions, e.g. by sequentially removing miRNA binding sites from Hox genes in their in-vitro system. I think this manuscript would greatly benefit from such an experimental validation that directly supports the general mechanism proposed by their theoretical modelling approach.

Minor points:

- 1) The authors demonstrate that miR-27 is required for the hysteresis of *Hoxa5* in response to varying levels of RA by exposing their EBs first to high RA and then to RARi (Figure 7F,G). However, in their characterization of the miR-27 mutant embryos they claim a significant caudal expansion of the *Hoxa5* only domain, suggesting that generally lower levels of RA are required to maintain *Hoxa5* (Figure 8G). How do the authors explain this discrepancy? Furthermore, would the outcome of the experiment be different if the authors instead of completely inhibiting RA signalling would expose their EBs to low levels of RA that are normally not sufficient to induce *Hoxa5* (i.e. is the RA concentration window over which hysteresis occurs just smaller than in wild-type)?
- 2) Related to the same figure, if I understand the schematics in Figure 7B correctly, the authors expose EBs first to defined concentrations of RA and then to RARi in combination with low or high RA in conditions 4 and 7? Is this correct? What is the rationale behind adding RA at the same time as RARi in these conditions?
- 3) For the miRNA overexpression experiments in Figure 9, it is not surprising that strong expression of a particular miRNA leads to a loss of the protein encoded by a mRNA targeted by the miRNA. To be sure, that these perturbations lead to a change in motor neuron identity, it is necessary to see whether the opposing Hox gene is upregulated under these conditions.
- 4) Figure 10G: The caudal expansion of the *Hoxa5* only expression domain (red) in the miR-27 double mutant seems to be exaggerated, given that the quantification of *Hoxa5* only cells at all axial identities suggests that their number is not significantly altered (Figure 10D). Instead, the major difference seems to be an increase in the number of *Hoxa5*/*Hoxc8*-coexpressing cells at the expense of *Hoxc8*-only cells.
- 5) Figure 6C,D: The authors should clearly mention in the figure caption that the different weights of the arrows depict the different degradation rate constants. Also, in the 2 schematics on the left of Figure 6D only one complex seems to have unbalanced degradation rates, so why do they fulfil the criteria for the emergence of bistability and negative feedback loops?
- 6) Figure 8A: I think it would help the reader to be consistent with the previous figures here and to depict the asymmetry of the degradation rate constants of the different miRNA-mRNA complexes in the model by varying the line weights of the arrows.

Reviewer #2:

In this work, Li and co-workers use a combination of experiments and mathematical modeling to investigate cell differentiation in the spinal cord. The authors find that the bistability needed to achieve stable cell fates is not due to a positive feedback loop at the level of transcriptional regulation but they rather claim that a new feedback mechanism based on interactions between microRNAs and mRNA underlies this phenomenon. They conclude that this feedback mechanism may be more common in tissue patterning.

How stable cell fates are controlled during cell differentiation is a fundamental question in developmental cell biology. Mathematical modeling and other theoretical approaches have played an important role in this field for decades and are becoming increasingly important. These complex phenomena can only be understood using a combination of theoretical and experimental approaches.

While the results of this work are potentially interesting, the manuscript in its current form is too long and difficult to follow. Most problematically, the essence of the mathematical models used and their underlying assumptions are not clearly described (see details below).

Major issues:

1. The main text describes many non-essential things in too much detail. For example, the models described at the beginning of the Results part (capturing positive feedback via transcriptional regulation and unilateral repression, respectively) do not need to be described at this level of detail: These are well-known results in the field that could be summarized very briefly. More generally, it may help to shorten the description of mathematical models that cannot explain the observed expression patterns. The number of different mathematical models used is excessive and will be hard to digest for most readers. A potential strategy could be to split the theoretical parts about the putative new mechanism of generating bistability and the experimental parts into two separate manuscripts.
2. Despite the length of this work, the form of the mathematical models and their key underlying assumptions do not become clear from the main text. For example, the model shown in Fig. 4A is reasonably complicated, involving eight molecular species and numerous positive and negative interactions. It is clear that the low-level details can only be shown comprehensively in the supplement, but it should still be explained in the main text what type of model is used (e.g. based on ODEs, capturing stochasticity or not, etc.) and how the regulatory and other interactions are captured (Hill functions or similar). Further, how many free parameters do these models have? How are the parameter values that apply to the real system estimated? How are general conclusions (independent of parameter values) drawn for these models? Given the length of the main text, it provides too little information about the mathematical models, which are at the heart of this work.
3. Closely related to the previous point, the most serious problem is that the new mechanism that produces bistability does not become completely clear. From lines 255-273, I understand that numerical simulations show bistability for some parameter values and the relation in Eq. (1) can be derived. However, it would be important to explain the mechanisms that underlies bistability in the model more clearly. Ideally, there should also be an intuitive explanation for this phenomenon (an attempt is made later in the main text but this did not make the essence of the putative mechanism clearer to me). Most important would be to explain the difference to other established mechanisms producing multi-stability beyond positive feedback loops (e.g. zero-order ultrasensitivity). The most convincing way to address this issue would be to first explain a minimal model (as simple as possible) where this putatively new phenomenon occurs in the main text, and then explain its connection to the more detailed models shown in Fig. 5. It would also be helpful to explain the origin of the (saddle-node?) bifurcation that leads to bistability in such a simplified model in the main text.
4. The detection limit and other potential limitations of the experimental methods used to determine mRNA and protein levels need to be discussed. This is important since the main conclusions about ruling out certain models depend critically on the co-occurrence (or lack thereof)

of specific mRNAs or proteins in cells. Co-occurrence might e.g. be observed for mRNAs but not for proteins if a more sensitive technique for detecting the former is used. This is not necessarily the case but this issue should be discussed and the key experimental conclusions corroborated by addressing these technical points.

5. In the last part (starting in line 480), it is claimed that experimental observations agree with predictions of the mmi-S model. It needs to be clarified if these are real predictions of the model, i.e. if they were made without fitting parameters and before obtaining the experimental data.

Reviewer #3:

In this work, Li et al used mathematical modeling, in vivo mouse embryo model studies to study cell fate decision during the process of spinal cord development. They first ruled out a series of possible regulation mechanisms between *Hoxa5* and *Hoxc8*, then proposed a feedback mechanism based on microRNA-mRNA interactions. Subsequent experimental studies further corroborate the model. This is a beautiful study showing how interplay between mathematical modeling and experiments lead to mechanistic understanding on cell fate decision during embryogenesis. The team has done a nice job.

Main comments

1) My main comment is that the mechanism of generating bistability through a miRNA-mRNA motif has been discussed in the literature, e.g., in the context of epithelial-to-mesenchymal transition (Tian et al. *Biophys J* 2013; Lu et al. *J Phys Chem B* 2013; Zhang et al. *Sci Sig* 2014). Specifically Tian et al. 2016 (which the authors cited) have thoroughly investigated a class of miRNA-mRNA modules, and showed a number of mechanisms that can generate bistability without apparent feedbacks. Some of the analyses discussed in this work, such as the effect of miRNA binding sites, have been systematically discussed in that paper. So the present work can be viewed as a biological example (or confirmation) of the predictions in that theoretical study. It might be appropriate to state it explicitly. For example, in lines 58-59 the authors state that "it is unclear if these interactions give rise to feedback loops with the capacity of bistable switches, particularly during mammalian development". It might be more accurate to say that while such mechanism of microRNA-mediated bistability have been predicted theoretically, there is no direct experimental demonstration, particularly during mammalian development.

2) I need explanation on equations 2a/2b in the supplemental material: why exponential form? Assume an exponential distribution of the morphogen along the RC axis? The authors need to justify how they approximate the 1D reaction-diffusion equation into these equations.

3) MicroRNAs usually have low specificity. Can existence of other targets of the microRNAs (acting as competitors) affect the network dynamics?

Minor comments

1) Fig. 1B: the authors may consider a different font selection for the caption of y axis.

2) Line 191 of supplemental: 10×40, not "X". Similar for a few other places.

Reviewer #1:

The establishment of different cell fates and sharp gene expression boundaries is critical for correct animal development. In this manuscript Li et al. utilize in-vivo developmental biology and stem-cell based in-vitro differentiation approaches in combination with computational modelling to achieve a mechanistic understanding of the underlying molecular principles by which Hoxa5-positive cervical and Hoxc8-positive brachial motor neurons arise in the developing spinal cord. Using immunofluorescent stainings for Hoxa5 and Hoxc8 they first study the protein dynamics that underlies the formation of this boundary in the developing mouse spinal cord. They then demonstrate that besides the formation of a sharp boundary at the protein level, the mRNAs of both Hox genes are co-expressed in a broad area of the developing spinal cord. This observation, in combination with previous observations that miRNAs are required for the correct expression of these Hox genes, led them to investigating the importance of post-transcriptional mechanisms in this process. Using computational modelling the authors demonstrate that 1) bi-stability can arise from mRNA-miRNA interactions if an mRNA harbours more than 1 binding site for a miRNA and the degradation rate constants follow a simple relationship and 2) that this bi-stability emerges as a consequence of negative feedback loops appearing due to the symmetries of the different configurations (unbound, bound by 1 miRNA, bound by 2 miRNAs, ...) being broken in opposite directions. They experimentally test if this mechanism can explain the formation of the Hoxa5 - Hoxc8 gene expression boundary by demonstrating that their model can account for the hysteresis in response to retinoic acid and the phenotypes observed upon different genetic manipulations in stem cell-derived motor neurons and the developing mouse embryo.

Overall this is a well-executed study that addresses an interesting topic - how different cell fates are robustly established during embryogenesis despite the presence of inherently noisy patterning systems. The authors combine computational modelling with experimental perturbations to demonstrate an important role of miRNAs during motor neuron subtype specification in the developing spinal cord and propose a conceptually novel mechanism how broadly expressed miRNAs can contribute to the formation of sharp gene expression boundaries. As miRNAs have important functions during all kind of processes ranging from development to tissue homeostasis and disease, I believe these findings will be interesting for a broad audience. While I am not the best person to judge the validity of the math underlying their modelling, the modelling section appears convincing and the authors clearly made an effort to explain their findings in a way that it can be understood by a broad audience. The experimental biology extends previous observations from the authors and other groups how miRNAs contribute to correct motor neuron patterning and subtype specification. The authors furthermore provide a high-quality single cell RNA sequencing dataset from embryonic motor neurons, which will be of broad interest to researchers working on spinal cord development or on the generation of motor neurons from stem cells, e.g. for therapeutic approaches. For these reasons I believe this manuscript will be a valuable addition to the field.

Response:

We are very grateful for the reviewer's thorough reading of our manuscript and the constructive comments and suggestions. We have performed additional experiments and analyses to address the points raised. In particular, we now provide new experimental evidence supporting the importance of multiple miRNA binding sites. We have also performed a new experiment to reconcile the seemingly contradictory results pointed out by the reviewer. We have made substantial changes to the manuscript to provide readers with more accurate interpretations of our results. We feel that our manuscript has been significantly improved by adopting the reviewer's suggestions.

1) The key outcomes of the modelling section are that for the emergence of bistability 1) multiple miRNA binding sites are required (lines 252-254, Figure 5A) and 2) the inhibitory processes for different miRNA-mRNA configurations are broken in opposite directions (lines 366 - 371, Figure 6D). Although the authors provide novel evidence that miR-27 and miR-196 interact with Hox genes and are required for correct motor neuron subtype specification, they never directly test these predictions, e.g. by sequentially removing miRNA binding sites from Hox genes in their in-vitro system. I think this manuscript would greatly benefit from such an experimental validation that directly supports the general mechanism proposed by their theoretical modelling approach.

Response:

This is an excellent suggestion. Experiments demonstrating the importance of multiple binding sites for miRNAs would provide strong support to the conclusions of this manuscript. To address the reviewer's point, we first mutated the seed sequences of multiple miRNA binding sites in the *Hoxc8* 3'UTR in a sequential manner to verify by luciferase assay that multiple miRNA binding sites are required for regulation of the *Hoxc8* 3'UTR (Figure R1, or new Figure EV4 A and B).

More specifically, we transfected a luciferase construct containing four conserved *miR-196* binding sites (predicted by TargetScan) in the *Hoxc8* 3' UTR into Hela cells. Upon overexpression of *miR-196*, we observed ~50% reduction in luciferase activity (Figure R1, or Figure EV4 B). To reduce the complexity, we selected the three most conserved canonical sites (site#1~3) for a mutation experiment, leaving site#4 intact in our experiment. We mutated the three binding sites individually, and then mutated two and three binding sites sequentially in *Hoxc8* 3' UTR reporter constructs (Figure R1, or Figure EV4 B). Interestingly, both site#1 single mutant and sites#2/3 double mutant manifested partially dampened responses to *miR-196* overexpression (Figure R1, or Figure EV4 B), suggesting a synergy between multiple sites. Furthermore, combined mutations in sites#1/2/3 triple mutant also led to reduced repression, which suggests that site #4 alone cannot achieve full repression. Thus, we have verified that multiple miRNA binding sites are required to exert the optimal targeting effect on *Hoxc8*.

Likewise, we used a similar strategy to perform luciferase experiment for *miR-27* in *Hoxa5* 3' UTR. Given that *miR-27* has only three conserved predicted targeting sites on *Hoxa5* 3' UTR, we mutated all individual and possible combinatorial sites (Figure EV4 C). Interestingly, we found that mutations in individual sites caused complete de-repression, which was also achieved with double or triple -site mutation (Figure R1, or Figure EV4 C). This result suggests

that *all three* binding sites are required for *miR-27* mediated repression of *Hoxa5*, a stronger form of cooperativity than the *miR-196* binding sites in the *Hoxc8* 3' UTR. This finding is also consistent with our previous study (Li *et al.*, 2017). Taken together, we have verified that multiple miRNA binding sites are required for both *Hoxc8* and *Hoxa5* in terms of the post-transcriptional regulation.

Figure R1 (included in Figure EV4). **Luciferase assay with 3' UTR.** (A) Predicted targeting sites for *miR-196* in the *Hoxc8* 3' UTR (left panel) and for *miR-27* in the *Hoxa5* 3' UTR (right panel), based on TargetScan. (B) left panel: Luciferase reporters were constructed with either a control *Hoxc8* 3' UTR or the 3' UTR sequence in which the individual or multiple potential target sites of *miR-196* were mutated (red). Right panel: Co-expression of a luciferase construct with *miR-196a* in HeLa cells silences a reporter carrying intact *miR-196* target sites, whereas *miR-196* fails to fully silence Mut1 (site#1 mutated), Mut2/3 (site#2 and site#3 mutated) and Mut1/2/3 (site#1, site#2, and site#3 mutated) luciferase constructs ($N=3$ independent experiments, mean \pm SD, * $P<0.05$, ** $P<0.01$). (C) left panel: Luciferase reporters were constructed with either a control *Hoxa5* 3'UTR or the 3' UTR sequence in which the individual or multiple potential target sites of *miR-27* were mutated (red). Right panel: Co-expression of a luciferase construct with *miR-27b* in ES cells silences a luciferase reporter constructs carrying one or more mutated sites ($N=3$ independent experiments, mean \pm SD, * $P<0.05$, ** $P<0.01$).

The description of these experiments has been added to the paragraph at the beginning of the section 'miRNA confers hysteresis in the response of *Hoxa5* to RA signaling' (Page 20, the second paragraph). Though these experiments do not directly demonstrate the importance of multiple binding sites in generating hysteresis, they provide additional support for the reaction network that

is used in our modeling work. Nevertheless, to examine the role of multiple binding sites more rigorously, the function of stoichiometric *Hoxa5/miR-27* interactions must be studied *in vivo*. A future study deploying conditional *miR-27* or *Hoxa5* 3'UTR knockout at the postmitotic stage is needed to dissect the role of *miR-27/Hoxa5* multiple targeting in *Hoxa5* biostability, which might be beyond the scope of the current manuscript.

Minor points:

1) The authors demonstrate that *miR-27* is required for the hysteresis of *Hoxa5* in response to varying levels of RA by exposing their EBs first to high RA and then to RARi (Figure 7F,G). However, in their characterization of the *miR-27* mutant embryos they claim a significant caudal expansion of the *Hoxa5* only domain, suggesting that generally lower levels of RA are required to maintain *Hoxa5* (Figure 8G). How do the authors explain this discrepancy? Furthermore, would the outcome of the experiment be different if the authors instead of completely inhibiting RA signalling would expose their EBs to low levels of RA that are normally not sufficient to induce *Hoxa5* (i.e. is the RA concentration window over which hysteresis occurs just smaller than in wild-type)?

Response:

The reviewer has made a sharp observation regarding the two results that seem to contradict each other, i.e. the *in vitro* experiments shown in original Figure 7F and G (revised Figure 6 F and G) suggesting that *miR-27* is required to maintain *Hoxa5* expression under the RA-free condition, yet the *in vivo* experiment shown in original Figure 8G (revised Figure 7G) suggests that *miR-27* suppresses *Hoxa5* expression in the caudal region where RA levels are low compared to the rostral region. To address this issue, we have now performed additional experiments (illustrated in new Figure 6E) and made slight modifications to our model to reconcile those results, as described in detail as follows (and shown in new Figure 6H):

The main goal of the *in vitro* experiments was to show that the system is bistable in the presence of *miR-27*, and is monostable in the absence of *miR-27*. Although this point was demonstrated successfully, it is still not trivial to relate this experiment to the knockout phenotype. We think that the RA-free (RA^{low} with RARi) condition we used previously in the ESC differentiation system might not have an *in vivo* counterpart near the *Hoxa5-Hoxc8* boundary, where significant levels of RA are likely present in embryos (see revised new model in Figure 6H). We have now performed a new experiment for EBs treated with an intermediate level of [RA]^{med} (250 nM). Under this condition, the inducible *miR-27* sponge (iMiR27-SP) cells showed a higher fraction of *Hoxa5*^{on} cells compared to control cells (new Figure 6F and G). Although this result is in direct contrast (increased *Hoxa5*^{on} cells) to our original result under the transient high-RA and final low-RA condition (decreased *Hoxa5*^{on} cells), it is consistent with our *in vivo* observations of the caudal region (new Figure 8). Importantly, both sets of results are consistent with our theoretical predictions, i.e. the control network has two stable states (*Hoxa5*^{on} and *Hoxa5*^{off}) in the low-RA

to medium-RA parameter region, whereas the *miR-27*-lacking network has a single stable state in which *Hoxa5* level increases monotonically with [RA]. Though this result is in agreement with our interpretations in the original manuscript, we have made a slight adjustment to our model to enhance the consistency between our model output and experimental data in terms of *Hoxa5* levels in the low-RA and medium-RA parameter regions (Figure R2). This adjustment does not affect any other result, including the spatiotemporal model that predicted the *in vivo* observations. The new and updated results are now shown in new Figure 6, and their descriptions are included in the updated text.

To report these new experiments and the revised model, we have added additional text to the first paragraph of Page 23 that describes new Figure 6.

Figure R2 (included in new Figure 6). Bifurcation analysis of a mathematical model based on our mmi-3 Model. Steady states of the system in the presence of *miR-27* (red, gray, and blue) or in the absence of *miR-27* (purple) at various concentrations of RA are shown. Solid curve: stable steady state. Dashed curve: unstable steady state (See Appendix Table S5 for parameter values).

Regarding the reviewer’s second question, we cannot exclude the possibility that the iMiR27-SP condition also exhibits bistability in an RA range narrower than the control. In the RA concentrations that we tested (0-500 nM), we did not observe any hysteresis effect, but it is possible that other molecular interactions may support some level of bistability. Testing complete loss of bistability in the iMiR27-SP condition would require higher detection sensitivity than that achievable with protein staining, as well as finer control of RA concentrations. We feel these experiments are beyond the scope of the current report. Nonetheless, we believe that our experiments have clearly demonstrated the critical impact of *miR-27* on hysteresis. To address the reviewer’s point, we have changed the word ‘necessary’ to ‘important’ in the first sentence of the first paragraph on Page 22.

2) Related to the same figure, if I understand the schematics in Figure 7B correctly, the authors expose EBs first to defined concentrations of RA and then to RARi in combination with low or high RA in conditions 4 and 7? Is this correct? What is the rationale behind adding RA at the same time as RARi in these conditions?

Response:

We agree with the reviewer that the original presentation was unclear. We added RARi for two primary reasons. Firstly, to prevent endogenous RA production from embryoid bodies (EBs), as the EBs also express small amounts of RALDH2 that might enable them to produce RAs via Vitamin A precursors in the medium (Peljto *et al*, 2010). Secondly, we wished to validate that the change is a specific reflection of RA signaling alone. We agree with the reviewer that adding a high concentration of RA together with RARi may not provide meaningful results. In fact, we have realized that the low-to-high dynamics of RA is not necessary for testing our hypothesis. Therefore, we have removed the last two conditions from this figure (old Figure 7B, revised Figure 6B). We have also added a brief explanation for our rationale of using RARi (Page 21, the second paragraph).

3) For the miRNA overexpression experiments in Figure 9, it is not surprising that strong expression of a particular miRNA leads to a loss of the protein encoded by a mRNA targeted by the miRNA. To be sure, that these perturbations lead to a change in motor neuron identity, it is necessary to see whether the opposing Hox gene is upregulated under these conditions.

Response:

We appreciate the reviewer's comment on this result. Unfortunately, the current differentiation protocol for generating rostral Hoxa5^{on} MNs (RA/Shh) and caudal Hoxc8^{on} MNs is relatively extreme in generating homogenous rostral or caudal populations (Mazzoni *et al*, 2013). It is technically challenging to obtain a consistent intermediate boundary region to reflect Hox identity in an ESC system as observable in embryos, thus making it difficult to obtain reciprocal Hox upregulation due to high RA (or FGF). In fact, we only used this system to perform overexpression as a proof-of-concept verification of the consequence of Hox reduction for miRNA overexpression. Future experiments to test this in a chicken *in ovo* system or using transgenic embryos might address this concern in a more convincing manner.

4) Figure 10G: The caudal expansion of the Hoxa5 only expression domain (red) in the miR-27 double mutant seems to be exaggerated, given that the quantification of Hoxa5 only cells at all axial identities suggests that their number is not significantly altered (Figure 10D). Instead, the major difference seems to be an increase in the number of Hoxa5/Hoxc8-coexpressing cells at the expense of Hoxc8-only cells.

Response:

We agree with the reviewer's criticism. We have now revised this diagram to indicate that we focus on the cervical C5~C7 segment, not the entire spinal cord. In addition, we have corrected the position of the co-expressing cells relative to the boundary in the new revised Figure 8G (old Figure 10G) to prevent confusion.

5) Figure 6C,D: The authors should clearly mention in the figure caption that the different weights of the arrows depict the different degradation rate constants. Also, in the 2 schematics on the left

of Figure 6D only one complex seems to have unbalanced degradation rates, so why do they fulfil the criteria for the emergence of bistability and negative feedback loops?

Response:

We have taken the reviewer's suggestion and now mention the arrow weights describing the differential degradation rate constants in the **legend of Figure 5** (old Figure 6).

With regard to Figure 5D, the reviewer has made a very good observation on a discrepancy between the mathematical requirement of bistability and our intuitive description of the requirement for a feedback loop. We have now supplemented our intuitive description with the following clarification:

As described in our original manuscript, for the double-arrow (2-binding-site) reaction network, both "symmetrical" (nondirectional) degradation rate constants (orange dot in Figure 5B) and "unidirectional asymmetry" (cyan line in Figure 5B) fail to give rise to bistability based on our mathematical analysis. Intuitively, we suggest that this is because both scenarios lack a clear underlying feedback loop. More precisely, each of the two scenarios has two opposing feedback loops that "cancel each other out". If we follow the dark blue arrows in each network of Figure 5D clockwise and then counterclockwise, the two resulting "loops" exhibit opposing directions and balanced strengths, thus producing no "net feedback". Only when the symmetry is broken such that the two underlying loops do not cancel each other out can the system have a net feedback loop. This requirement can be satisfied by the scenario with only one complex having unbalanced degradation rates (Figure 5D, two bottom networks - note that we have changed the positions of the networks for easier description), as pointed out by the reviewer. This net feedback explanation also encompasses other scenarios described in Figure 5D.

We have now incorporated this explanation in the **sixth subsection of our Results (Pages 18-19)**. We have also updated **Figure 5C and D** to reflect these changes.

6) Figure 8A: I think it would help the reader to be consistent with the previous figures here and to depict the asymmetry of the degradation rate constants of the different miRNA-mRNA complexes in the model by varying the line weights of the arrows.

Response:

The reviewer has made a very good suggestion here. We have now adjusted the weights of the degradation arrows in **Figure 7A (old Figure 8A)** to reflect the asymmetry of the degradation rate constants. These constants are those reflected in the mmi-S Model.

It should be noted that a key theoretical result of this study is that the degradation rate constants can be very flexible as far as the bistability is concerned (Figure 5D), so there exists many distinct and valid ways to assign the weights of the arrows in Figure 7A. Given that experimental estimation of these rate constants remains challenging, it is not our intention to suggest specific values for the *Hoxa5*-miR-27 and *Hoxc8*-miR-196 systems in this manuscript. Nonetheless, we do raise a preferred hypothesis about these rate constants. Since the *miR-27* binding sites on *Hoxa5*

mRNA do not satisfy the requirement for target-mediated miRNA degradation (TDMD), which usually involves extended base-pairing at the 3' direction of the binding site (Ghini *et al*, 2018; Marcinowski *et al*, 2012), we used a parameter set that has cooperative mRNA degradation but not TDMD. In contrast, some of the *miR-196* binding sites on *Hoxc8* mRNA satisfy the sequence requirement of TDMD (Figure EV 4), so we used a parameter set that involved TDMD-driven bistability.

Reviewer #2:

In this work, Li and co-workers use a combination of experiments and mathematical modeling to investigate cell differentiation in the spinal cord. The authors find that the bistability needed to achieve stable cell fates is not due to a positive feedback loop at the level of transcriptional regulation but they rather claim that a new feedback mechanism based on interactions between microRNAs and mRNA underlies this phenomenon. They conclude that this feedback mechanism may be more common in tissue patterning.

How stable cell fates are controlled during cell differentiation is a fundamental question in developmental cell biology. Mathematical modeling and other theoretical approaches have played an important role in this field for decades and are becoming increasingly important. These complex phenomena can only be understood using a combination of theoretical and experimental approaches.

While the results of this work are potentially interesting, the manuscript in its current form is too long and difficult to follow. Most problematically, the essence of the mathematical models used and their underlying assumptions are not clearly described (see details below).

Response:

We thank the reviewer for very helpful suggestions to improve the clarity of the manuscript. We agree with the comment about the problems regarding model descriptions in the original version. We have now made substantial changes to the manuscript so that the less important models are now presented in a more concise way, with the important minimal model and the underpinned bistability mechanism being described in greater detail. We also present very interesting observations regarding the diverse forms of system nonlinearity. Furthermore, we now compare the new mechanism with existing literature on other mechanisms of bistability based on the reviewer's suggestion.

Major issues:

1. The main text describes many non-essential things in too much detail. For example, the models described at the beginning of the Results part (capturing positive feedback via transcriptional regulation and unilateral repression, respectively) do not need to be described at this level of detail: These are well-known results in the field that could be summarized very briefly. More generally, it may help to shorten the description of mathematical models that cannot explain the observed

expression patterns. The number of different mathematical models used is excessive and will be hard to digest for most readers. A potential strategy could be to split the theoretical parts about the putative new mechanism of generating bistability and the experimental parts into two separate manuscripts.

Response:

We agree with the reviewer's comments. We have now moved the figure panels of the T-CR model to Expanded View (Figure EV2). We have also removed the text describing that model from the second subsection of our Results (Page 7). We considered splitting the manuscript into two, but we feel that there are advantages to having all of the results in a single manuscript since it is an authentic reflection of the integrated experimental-modeling approach employed in this project. Importantly, the Editor has also recommended not splitting the manuscript. We have further shortened some parts of the Introduction and Discussion to make the manuscript more concise.

Note, we have also adopted this reviewer's suggestion in Comment #2 to provide more details about our models. We think that some of these details needed to be introduced at the stage of the first model description, and this is now done in the second paragraph of the second subsection of our Results (Page 7).

2. Despite the length of this work, the form of the mathematical models and their key underlying assumptions do not become clear from the main text. For example, the model shown in Fig. 4A is reasonably complicated, involving eight molecular species and numerous positive and negative interactions. It is clear that the low-level details can only be shown comprehensively in the supplement, but it should still be explained in the main text what type of model is used (e.g. based on ODEs, capturing stochasticity or not, etc.) and how the regulatory and other interactions are captured (Hill functions or similar). Further, how many free parameters do these models have? How are the parameter values that apply to the real system estimated? How are general conclusions (independent of parameter values) drawn for these models? Given the length of the main text, it provides too little information about the mathematical models, which are at the heart of this work.

Response:

We agree with the reviewer's comments. We now provide more necessary information on these models. Including the model types (ODE), we also provide descriptions for morphogen noise (white noise terms), the nonlinear functions for transcriptional regulation (Hill function) and miRNA regulation (mass action kinetics), the number of free parameters (see below), and the way we sampled the parameter values. Similar to other models for mammalian tissues (Balaskas *et al*, 2012; Zagorski *et al*, 2017), we were not able to estimate the parameter values based on experimental data. This is in part due to the unavailability of absolute concentrations of key molecules in the system, including morphogens, *Hox* mRNAs, *Hox* proteins, and miRNAs. To show that our conclusions do not depend on the choice of particular parameter values, we adopted an approach that is widely used in similar modeling studies (e.g. (Zagorski *et al.*, 2017)), whereby

values of key parameters were randomly drawn from log-normal distributions (Appendix Table S4) that cover a wide range of biologically plausible behaviors. The range for the threshold parameter (K) of activation/inhibition Hill functions was chosen such that the activation/inhibition can occur at low, intermediate, or high concentrations of the regulator. For each model, 10,000 parameter sets were randomly chosen, and model performance was evaluated for each parameter set. Equivalent parameters in multiple models have the same ranges of values. These results are summarized in Figure EV3 G. Our conclusions are based on the evaluations of all tested parameter sets rather than a few representative parameter sets.

We took the reviewer's suggestion and included some additional detail for the models in various locations of the revised manuscript. In the second paragraph of the second subsection of the Results (Page 7), we have added:

‘Next, we built two ordinary differential equation (ODE) models based on for the T-CR network and a Transcriptional Unilateral Repression (T-UR) network respectively (Figure EV2). Transcriptional regulation and cooperativities were described by Hill functions (Appendix Supplementary Methods). We found that the T-CR Model exhibited desirable performance in lineage segregation in the presence of noisy morphogen signals described by white noise terms in the ODEs (Figure EV2 B-D, Movie S1) We found that the T-CR Model had desirable performance in lineage segregation in the presence of noisy morphogen signals described by white noise terms in the ODEs (Figure EV2 B-D, Movie S1), whereas the T-UR Model produced fluctuating cell lineages and a blurred tissue boundary under the same condition despite assuming very high cooperativity of gene regulation (Figure EV2 E-G and Movie S2).’

In the first paragraph of the fourth subsection of the Results (Page 10), we have added:

‘This new model describes miRNA-mediated mRNA degradation and inhibition, with mass action kinetics similar to previously established approaches (Lu *et al*, 2013b; Osella *et al*, 2011) and transcriptional regulation having the same descriptors as for the T-CR and T-UR models. With a representative parameter set, ...’

In the last paragraph of the same subsection (Page 11), we have added:

‘To exclude the possibility that our conclusions on model performance are sensitive to the choice of parameter values, we performed random sampling of parameter values over a wide and biologically plausible range for the T-CR, T-UR, Tmi-UR and Tmi-FB Models (Appendix Table S4), which have 8, 7, 17 or 19 varied parameters, respectively. For each of these models, we randomly chose 10,000 parameter sets and performed simulations using the same procedure.’

We believe that including this new content in the main text renders the presentation of our models clearer to readers.

3. Closely related to the previous point, the most serious problem is that the new mechanism that produces bistability does not become completely clear. From lines 255-273, I understand that numerical simulations show bistability for some parameter values and the relation in Eq. (1) can

be derived. However, it would be important to explain the mechanisms that underlies bistability in the model more clearly. Ideally, there should also be an intuitive explanation for this phenomenon (an attempt is made later in the main text but this did not make the essence of the putative mechanism clearer to me). Most important would be to explain the difference to other established mechanisms producing multi-stability beyond positive feedback loops (e.g. zero-order ultrasensitivity). The most convincing way to address this issue would be to first explain a minimal model (as simple as possible) where this putatively new phenomenon occurs in the main text, and then explain its connection it to the more detailed models shown in Fig. 5. It would also be helpful to explain the origin of the (saddle-node?) bifurcation that leads to bistability in such a simplified model in the main text.

Response:

We agree with the reviewer's criticism. In our original manuscript, we focused on providing an intuitive explanation for the positive feedback loop, but we did not sufficiently explain the source of bistability in terms of system nonlinearity, which is also essential for bistability. We now use the mmi-2 Model to address this issue. This model is considered minimal in our study because we have shown that the mmi-1 Model does not generate bistability (Figure 5A, Appendix Supplementary Methods 1.7). The mmi-2 Model is now described in a 2-ODE 2-algebraic-equation (AE) system (Figure 4B, Eq A, or a slightly simpler form in Appendix Supplementary Methods Eq 1.8.4). We have revised the equations in Figure 4B to reflect the minimal form of the mmi-2 Model. Although it is possible to further reduce the number of equations by eliminating one of the AEs, we found that this would cause loss of intuition and not much mathematical convenience would be gained.

We have now overcome some technical challenges and generated two phase planes to explain the source of bistability in three representative scenarios (Figure 4D, or Figure R3). The three phase planes correspond to three biologically plausible mechanisms: cooperativity of two binding sites for triggered degradation rates of mRNA, cooperativity in miRNA degradation, and target-induced miRNA degradation (TDMD) that occurs when mRNA concentration is higher than miRNA concentration. All phase planes show that the bistability arises from nonlinear relationships between mRNA and miRNA in terms of their differential degradation rates in the complexes. Very interestingly, given the broad range of bistability-enabling parameters (Eq 1), the sources of nonlinearity are diverse. For example, in the TDMD mechanism, it is clear that the two sigmodal nullclines give rise to three intersections (two stable nodes and one saddle point), and that the shapes of these nullclines are due to the triggered degradation rates of mRNA and miRNA that occur in different concentration regions, which give rise to different thresholds (Figure R3, right). The cooperative scenario is less intuitive, but it is not surprising to see the Z-shaped response curve as one of the nullclines (Figure R3, left, for example), given the nonlinear form of the equations. In fact, this is a common response curve when a system has a positive feedback loop (Novak & Tyson, 1993). Interestingly, this nullcline allows bistability even if the other response curve (miRNA as a function of mRNA) is 'flat'. Overall, we believe that the new representative phase planes make the source of bistability clearer to readers.

Figure R3 (included in new Figure 4). **Phase planes for the mmi-2 Model** (Eq A in Figure 4B). Three representative parameter sets were chosen. Left: cooperative mRNA degradation ($10a_1 = a_2 = 10, b_1 = b_2 = 1$). Middle: cooperative miRNA degradation ($a_1 = a_2 = 1, b_1 = 10b_2 = 1$). Right: target-mediated miRNA degradation (TDMD) ($2a_1 = a_2 = 2, b_1 = 2b_2 = 2$). All other parameters are equal to 1, except $Kk_R^0/s_R = 10^{-5}$. Dark red and black: stable steady states. Light blue: unstable steady state. Blue curve: nullcline of ODE for R_T . Orange curve: nullcline of ODE for r_T (Eq A in Figure 4B). Gray arrows: representative directions in the vector field. Gray curve: separatrix. To obtain the nullclines vector field, coordinates in the R_T, r_T space were used as the initial conditions for ODEs in Eq A. The corresponding values of C_1 and C_2 were obtained by solving the algebraic equations in Eq A with the initial conditions. Resulting values for the right-hand side of the ODEs were used to determine the positions of the nullclines.

We also agree with the reviewer’s suggestion to clarify the difference between the source of bistability in our model and previously published ones. Bistability-enabling nonlinearity often arises from cooperativity in binding affinities (for instance, a transcription factor that binds to multiple sites), which is described by Hill functions (Ferrell Jr & Ha, 2014). Our model does not contain such a mechanism (the dissociation constants of mRNA and miRNA are identical for all binding sites) or any nonlinear functions similar to Hill functions. As the reviewer has pointed out, nonlinearity may also arise from zero-order ultrasensitivity, which depends on particular types of enzymatic reactions in which the low-concentration enzymes are saturated by the substrates near the threshold of the responses (Goldbeter & Koshland, 1981). Our model does not contain Michaelis-Menten kinetics, and the concentrations of mRNA and miRNA in our model are comparable. Another bistability mechanism involves chained reactions that share enzymes (Markevich *et al*, 2004), which also requires Michaelis-Menten kinetics. Our model does not involve such a mechanism.

To address the reviewer’s comment, we have included the new phase planes in Figure 4, and have added the following paragraph to the subsection ‘Elementary mRNA and miRNA interactions with simple kinetic requirements can generate bistability’ (Page 15):

‘We illustrate the sources of bistability with phase planes for three representative parameter sets corresponding to cooperative mRNA degradation, cooperative miRNA degradation and TDMD, respectively (Figure 4D). In the first scenario (Figure 4D left), cooperative mRNA degradation from binding with multiple sites gave rise to a Z-shaped response curve (nullcline) governed by the ODE for the total mRNA (Figure 4B, Eq A). In the second scenario (Figure 4D middle), a

nonlinear steady-state miRNA level emerged in response to the change in mRNA level. In the third scenario (Figure 4D, right), sigmoidal nullclines for ODEs of both total mRNA and total miRNA were generated from the triggered degradation of both molecules. In each scenario, nonlinear relationships between the two variables gave rise to two stable steady states (nodes, Figure 4D, red and black), and one unstable steady state (saddle point, Figure 4D, light blue). Changes in parameter values results in coalescence and disappearance of a saddle point and a node, which underlie the saddle-node bifurcations observed for the bistable switches (Figure 4C, both ends of blue curves). Notably, the sources of the nonlinearity in our model differ from several well-known mechanisms crucial for bistability, including cooperativity in binding affinities (Hill functions) (Balaskas *et al.*, 2012), saturable enzymes for covalent modifications (zero-order ultrasensitivity) (Goldbeter & Koshland, 1981), and shared enzymes for multiple reactions (Markevich *et al.*, 2004). Our model involves the law of mass action for elementary reactions, but not Michaelis-Menten kinetics or any phenomenological nonlinear function.’

4. The detection limit and other potential limitations of the experimental methods used to determine mRNA and protein levels need to be discussed. This is important since the main conclusions about ruling out certain models depend critically on the co-occurrence (or lack thereof) of specific mRNAs or proteins in cells. Co-occurrence might e.g. be observed for mRNAs but not for proteins if a more sensitive technique for detecting the former is used. This is not necessarily the case but this issue should be discussed and the key experimental conclusions corroborated by addressing these technical points.

Response:

The reviewer has raised a good point. The detection limits of protein staining might be a factor contributing to our conclusion that the distributions of the two Hox proteins do not overlap at E12.5. We now discuss this point in the revised manuscript. However, the limits of mRNA detection would not have affected our conclusion. The technique of RNA-sequencing is highly specific in terms of detected transcripts, so co-expression of *Hoxa5* and *Hoxc8* mRNAs in those ‘double-positive’ cells in Figure 3D and E mostly likely represents true positives. Indeed, the possible detection limits of RNA-sequencing means that there might be even more *Hoxa5-Hoxc8* co-expressing cells near the boundary. Although the detection limits of protein staining could influence our conclusions, we would like to point out that our conclusions are not based solely on the staining levels in single cells. The clear boundary formation (Figure EV1) is also a strong indication that lineage segregation is complete at the tissue level. Therefore, even if some *Hoxa5* and *Hoxc8* proteins are expressed at low levels on the incorrect sides of the boundary, this segregation would still be in stark contrast to the mRNA distribution (Figure 3E) that displays no evidence of bimodal $Hoxa5^{high}Hoxc8^{low}$ and $Hoxa5^{high}Hoxc8^{low}$ patterns with any arbitrary cutoffs. To address this point, we have now added the following text to the last subsection of our Discussion (Page 30):

‘We observed that lineage segregation at the tissue boundary is manifested by proteins rather than mRNAs. Although the detection limits of protein staining may contribute to the absence of *Hoxa5*

or Hoxc8 signals in some cells, the clearly segregated distributions of these two proteins at E12.5 (Figure EV1), together with upregulation of the downstream factors (Figure 3D), support the lineage commitment of almost all MNs and the formation of a functional boundary. This scenario is in stark contrast to mRNA distributions that do not display any pattern of segregation (Figure 3).'

5. In the last part (starting in line 480), it is claimed that experimental observations agree with predictions of the mmi-S model. It needs to be clarified if these are real predictions of the model, i.e. if they were made without fitting parameters and before obtaining the experimental data.

Response:

We agree with the reviewer that we should clarify how we obtained the parameter values for the mmi-S Model. The model was fit to known observations on the distributions of *Hox* mRNAs, proteins, and miRNAs. The predictions were made without fitting to the *in vivo* experimental data presented in the last subsection of our Results. The directions of the boundary shifts are quite intuitive from the structure of the model. To address this point, we have added the following text to the first paragraph of the second-last subsection of our Results (Page 23):

'We fitted this model to observed distributions of *Hox* mRNA and proteins (Figures 1 and 3) (Li *et al.*, 2017; Wong *et al.*, 2015). The ranges of the parameters were the same as those used for all other models (Figure EV3 I), except for the newly identified feedback mediated by miRNA.'

Reviewer #3:

In this work, Li et al used mathematical modeling, in vivo mouse embryo model studies to study cell fate decision during the process of spinal cord development. They first ruled out a series of possible regulation mechanisms between Hoxa5 and Hoxc8, then proposed a feedback mechanism based on microRNA-mRNA interactions. Subsequent experimental studies further corroborate the model. This is a beautiful study showing how interplay between mathematical modeling and experiments lead to mechanistic understanding on cell fate decision during embryogenesis. The team has done a nice job.

Response:

We thank the reviewer for his/her enthusiasm for our work and for his/her constructive suggestions. We now discuss the relevance of our work with regard to several important previous publications highlighted by the reviewer. In addition, we have provided a detailed derivation of the exponential distributions of morphogen signals. We have also performed additional simulations and now discuss the effect of nonspecific miRNA binding in the revised manuscript.

Main comments

1) My main comment is that the mechanism of generating bistability through a miRNA-mRNA motif has been discussed in the literature, e.g., in the context of epithelial-to-mesenchymal transition (Tian et al. Biophys J 2013; Lu et al. J Phys Chem B 2013; Zhang et al. Sci Sig 2014). Specifically Tian et al. 2016 (which the authors cited) have thoroughly investigated a class of miRNA-mRNA modules, and showed a number of mechanisms that can generate bistability without apparent feedbacks. Some of the analyses discussed in this work, such as the effect of miRNA binding sites, have been systematically discussed in that paper. So the present work can be viewed as a biological example (or confirmation) of the predictions in that theoretical study. It might be appropriate to state it explicitly. For example, in lines 58-59 the authors state that "it is unclear if these interactions give rise to feedback loops with the capacity of bistable switches, particularly during mammalian development". It might be more accurate to say that while such mechanism of microRNA-mediated bistability have been predicted theoretically, there is no direct experimental demonstration, particularly during mammalian development.

Response:

We have adopted the reviewer's suggestion and now discuss the current literature more broadly and more clearly. We also now cite publications relating to miRNA-mediated feedback loops involving transcriptional inhibition at the beginning of the second paragraph of the fourth subsection of our Results (Page 11):

‘Previous studies on other systems have reported the possibility of a functional TF-miRNA feedback loop whereby the TF transcriptionally inhibits the miRNA that, in turn, represses the TF (Celià-Terrassa *et al.*, 2018; Lu *et al.*, 2013a; Palm *et al.*, 2013; Tian *et al.*, 2013; Tsang *et al.*, 2007; Yoon *et al.*, 2011; Zhang *et al.*, 2014)’

In addition, we have now highlighted prior theoretical work on miRNA-mRNA-mediated bistability *directly before* we describe our key theoretical findings as follows ‘a miRNA model proposed previously (Tian *et al.*, 2016), with this latter showing bistability arising from interactions between miRNA and mRNA with multiple binding sites in the absence of transcriptional inhibition’ (Page 13). Thus, we now clearly show that our work is inspired by earlier research. We have decided to remove the description of miRNA from the first paragraph of our Introduction because it seemed to disrupt the flow of the first and second paragraphs, both of which describe the widely accepted mechanism of transcriptional feedback loop.

While we appreciate the pioneering work mentioned by the reviewer and we agree that previous publications have shown that bistability can arise from miRNA-mediated reactions without an explicit feedback loop, we believe that our work provides new exciting theoretical insights into this topic. Mathematically, we have used an innovative approach to show the precise parameter range for bistability (Eq 1; 1.8.2 in Appendix Supplementary Methods), and that this range is significantly wide given the biological constraints. Conceptually, we have shown the source of a hidden feedback loop arising from the model. To the best of our knowledge, these potentially important pieces of information are not available from any previous works. Therefore, we do not think that our work represents an experimental confirmation of previous theoretical work.

2) I need explanation on equations 2a/2b in the supplemental material: why exponential form? Assume an exponential distribution of the morphogen along the RC axis? The authors need to justify how they approximate the 1D reaction-diffusion equation into these equations.

Response:

We agree with the reviewer that our reason for using the exponential function to describe morphogen distributions was not clear in the original manuscript. To clarify this point, we now provide detailed derivations of this function. We have included the key steps in the Appendix Supplementary Methods (Page 5), as well as below in this response letter:

We consider a one-dimensional tissue domain with a length of L , and a reaction-diffusion equation describing a morphogen concentration M that is a function of time and space.

$$\frac{dM}{dt} = D \frac{d^2M}{dx^2} - kM$$

Here, D is the diffusion coefficient (length squared per unit time). k is the degradation rate constant (the reciprocal of unit time). As a biological constraint, we assume that there is a source of morphogen production at the boundary $x = 0$, and that the boundary $x = L$ has a relatively low concentration of the morphogen.

At the steady state, the distribution M_s is described by:

$$0 = D \frac{d^2 M_s}{dx^2} - k M_s$$

Suppose $u(x)$ is the solution to this second-order ODE, we have the general solution:

$$u(x) = A e^x + B e^{-x}$$

where A and B are constants that depend on the boundary conditions. Since the solution should hold for tissues that are very long relative to the size of a cell ($L \rightarrow \infty$), we must have $A = 0$. If $A \neq 0$, then the e^x term will diverge to infinity as $x \rightarrow \infty$, while e^{-x} approaches zero, and the solution will be inconsistent with the biological constraint.

For boundary conditions, we assume constant flux at one end of the tissue and a constant low concentration at the other end:

$$\frac{dM(0, t)}{dx} = \sigma, \quad t \geq 0$$

$$M(L, t) = \varepsilon, \quad t \geq 0$$

Given these boundary conditions, we obtain:

$$B = \sigma = \varepsilon e^L$$

Therefore, the steady state distribution of the morphogen is:

$$M_s = u(x) = \sigma e^{-x}$$

In addition to this mathematical derivation, we are also aware that previous reports present similar assumptions about morphogen distributions (Lander, 2013; McHale *et al*, 2006). This is a common approximation because the exponential form is derived from very basic assumptions. We have now included these results in Section 1.2 of the Appendix Supplementary Methods (Page 5).

3) MicroRNAs usually have low specificity. Can existence of other targets of the microRNAs (acting as competitors) affect the network dynamics?

Response:

This is a very good question. miRNAs usually have multiple target genes. Whether the existence of competitors can affect the dynamics depends on a few factors, some of which are difficult to estimate with the existing experimental data. Here, we discuss some of these factors and their possible impact on the bistability arising from the network, using a newly constructed mathematical model stemming from the mmi-2 Model.

In addition to a miRNA-target pair (R and r), we considered a hypothetical competitor mRNA R_c that can bind to miRNA (r). We found that the competitor miRNA can affect the behavior of the bistable switch. However, the effect depends on at least four factors: 1) the relative abundance of R and R_c ; 2) the number of binding sites on R_c ; 3) the altered degradation rate constants upon

binding of R_c and miRNA; and 4) the dissociation constant of the binding between R and r, and that between R_c and r. We focused our discussion on the ability of the system to have a clear bistable switch for the primary target mRNA R because this bistable switch would allow us to find parameters for the mmi-S Model that explain the experimental data. First, we considered two scenarios (R_c has two binding sites or it has one binding site). If R_c has two binding sites and the triggered degradation rate constants satisfy Eq 1 ($2a_1/b_1 < a_2/b_2$), then the system is bistable for both R and R_c. This is an obvious scenario because the variable R in the original mmi-2 Model can be viewed as the total concentration of multiple targets with similar properties. Therefore, we focused on the situation where R_c cannot generate bistability by itself, and the triggered degradation rate constants follow the average experimentally estimated quantities ($2a_1 = a_2 = 2$, $b_1 = b_2 = 1$) (Grimson *et al*, 2007). The degradation rate constants for R are assumed to be moderately cooperative ($3a_1 = a_2 = 3$), which gives rise to a small bistable region (Figure R4A, top left). We varied the transcription rate constants for R_c, as well as the relative dissociation constants of the primary target and the competitor mRNA, and performed bifurcation analysis for 15 cases (Figure R4A). As expected, when target-miRNA binding is 10,000 times stronger than the competitor-miRNA binding, the system showed a clear on-off bistable switch for R (Figure R4A, top row), and when target-miRNA binding is 10,000 times weaker than the competitor-miRNA binding, the bistable switch disappeared (Figure R4A, bottom row). Very interestingly, in most other cases, the clear on-off bistable switches were retained (Figure R4A), including a situation where the target-miRNA binding is 100 times weaker than the competitor-miRNA binding. The only parameter region that resulted in the disappearance of the bistable switches is when the competitor concentration is at least comparable to the target concentration and the target-miRNA binding is 100 times stronger than the competitor-miRNA binding. We found that the overall effects of a single-binding-site competitor were similar to the double-binding-site competitor that is not bistability enabling (Figure R4B). In conclusion, the competitor mRNA may have a negative impact on the bistable switch mechanism proposed in this study, but in a wide range of biologically plausible parameter values, the bistability was still retained.

To address the reviewer's question, we have now included this discussion in Appendix Supplementary Methods 1.12 (Page 25), and incorporated the new figure as Appendix Figure S5.

Figure R4 (included in new Appendix Figure S5). **Bifurcation diagrams of the mmi-S Model in the presence of a competitor mRNA.** The mmi-S Model was combined with a set of equations describing a competitor mRNA R_c , which shares the same regulatory miRNA with R (a primary target considered in

this study). Red curve: stable steady state with high amounts of free primary target mRNA. Black curve: stable steady state with low amounts of free primary target mRNA. **A.** The competitor mRNA has two binding sites. The degradation rate constants are additive, but not cooperative ($2a_1 = a_2 = 2$), which does not allow it to generate a bistable switch by itself (right). **B.** The competitor mRNA has one binding site. Light blue curve: unstable steady state of free primary target mRNA. No TDMD was considered for this competition model. Orange curve: steady state of free competitor mRNA (Solid: stable. Dashed: unstable). Scaled dissociation constants (Kk_R^0/s_R , Kk_{RC}^0/s_{RC}) were varied for plots from top to bottom with the indicated values. Transcription rate constants (s_{RC}) of the competitor mRNA were varied for plots from left to right with the indicated values. All other parameters, except the control parameter for bifurcation analysis, were set to 1.

Minor comments

1) Fig. 1B: the authors may consider a different font selection for the caption of y axis.

Response:

We have now used a different font that is more consistent with other labels in our figures.

2) Line 191 of supplemental: 10×40, not "X". Similar for a few other places.

Response:

Well spotted. We have now corrected all instances of this error in the Appendix Supplementary Methods and figure legends.

References

- Balaskas N, Ribeiro A, Panovska J, Dessaud E, Sasai N, Page KM, Briscoe J, Ribes V (2012) Gene regulatory logic for reading the Sonic Hedgehog signaling gradient in the vertebrate neural tube. *Cell* 148: 273-284
- Celià-Terrassa T, Bastian C, Liu D, Ell B, Aiello NM, Wei Y, Zamalloa J, Blanco AM, Hang X, Kunisky D *et al* (2018) Hysteresis control of epithelial-mesenchymal transition dynamics conveys a distinct program with enhanced metastatic ability. *Nat Commun* 9: 5005
- Ferrell Jr JE, Ha SH (2014) Ultrasensitivity part I: Michaelian responses and zero-order ultrasensitivity. *Trends Biochem Sci* 39: 496-503
- Ghini F, Rubolino C, Climent M, Simeone I, Marzi MJ, Nicassio F (2018) Endogenous transcripts control miRNA levels and activity in mammalian cells by target-directed miRNA degradation. *Nat Commun* 9: 3119
- Goldbeter A, Koshland DE (1981) An amplified sensitivity arising from covalent modification in biological systems. *Proc Natl Acad Sci USA* 78: 6840-6844
- Grimson A, Farh KK-H, Johnston WK, Garrett-Engele P, Lim LP, Bartel DP (2007) MicroRNA targeting specificity in mammals: determinants beyond seed pairing. *Mol Cell* 27: 91-105
- Lander AD (2013) How cells know where they are. *Science* 339: 923-927
- Li C-J, Hong T, Tung Y-T, Yen Y-P, Hsu H-C, Lu Y-L, Chang M, Nie Q, Chen J-A (2017) MicroRNA filters Hox temporal transcription noise to confer boundary formation in the spinal cord. *Nat Commun* 8: 14685
- Lu M, Jolly MK, Gomoto R, Huang B, Onuchic J, Ben-Jacob E (2013a) Tristability in cancer-associated microRNA-TF chimera toggle switch. *The journal of physical chemistry B* 117: 13164-13174
- Lu M, Jolly MK, Levine H, Onuchic JN, Ben-Jacob E (2013b) MicroRNA-based regulation of epithelial-hybrid-mesenchymal fate determination. *Proc Natl Acad Sci U S A*
- Marcinowski L, Tanguy M, Krmpotic A, Rädle B, Lisnić VJ, Tuddenham L, Chane-Woon-Ming B, Ruzsics Z, Erhard F, Benkartek C (2012) Degradation of cellular mir-27 by a novel, highly abundant viral transcript is important for efficient virus replication in vivo. *PLoS Pathog* 8: e1002510
- Markevich NI, Hoek JB, Kholodenko BN (2004) Signaling switches and bistability arising from multisite phosphorylation in protein kinase cascades. *The Journal of Cell Biology* 164: 353
- Mazzoni EO, Mahony S, Peljto M, Patel T, Thornton SR, McCuine S, Reeder C, Boyer LA, Young RA, Gifford DK (2013) Saltatory remodeling of Hox chromatin in response to rostrocaudal patterning signals. *Nat Neurosci* 16: 1191-1198
- McHale P, Rappel W-J, Levine H (2006) Embryonic pattern scaling achieved by oppositely directed morphogen gradients. *Phys Biol* 3: 107
- Novak B, Tyson JJ (1993) Numerical analysis of a comprehensive model of M-phase control in *Xenopus* oocyte extracts and intact embryos. *J Cell Sci* 106: 1153-1168

- Osella M, Bosia C, Corá D, Caselle M (2011) The role of incoherent microRNA-mediated feedforward loops in noise buffering. *PLoS Comput Biol* 7: e1001101
- Palm T, Hemmer K, Winter J, Fricke IB, Tarbashevich K, Sadeghi Shakib F, Rudolph I-M, Hillje A-L, De Luca P, Bahnassawy La (2013) A systemic transcriptome analysis reveals the regulation of neural stem cell maintenance by an E2F1–miRNA feedback loop. *Nucleic Acids Res* 41: 3699-3712
- Peljto M, Dasen JS, Mazzoni EO, Jessell TM, Wichterle H (2010) Functional diversity of ESC-derived motor neuron subtypes revealed through intraspinal transplantation. *Cell Stem Cell* 7: 355-366
- Tian X-J, Zhang H, Xing J (2013) Coupled reversible and irreversible bistable switches underlying TGF β -induced epithelial to mesenchymal transition. *Biophys J* 105: 1079-1089
- Tian XJ, Zhang H, Zhang J, Xing J (2016) Reciprocal regulation between mRNA and microRNA enables a bistable switch that directs cell fate decisions. *FEBS Lett* 590: 3443-3455
- Tsang J, Zhu J, Van Oudenaarden A (2007) MicroRNA-mediated feedback and feedforward loops are recurrent network motifs in mammals. *Mol Cell* 26: 753-767
- Wong SFL, Agarwal V, Mansfield JH, Denans N, Schwartz MG, Prosser HM, Pourquié O, Bartel DP, Tabin CJ, McGlenn E (2015) Independent regulation of vertebral number and vertebral identity by microRNA-196 paralogs. *Proc Natl Acad Sci USA* 112: E4884-E4893
- Yoon WH, Meinhardt H, Montell DJ (2011) miRNA-mediated feedback inhibition of JAK/STAT morphogen signalling establishes a cell fate threshold. *Nat Cell Biol* 13: 1062-1069
- Zagorski M, Tabata Y, Brandenberg N, Lutolf MP, Tkačik G, Bollenbach T, Briscoe J, Kicheva A (2017) Decoding of position in the developing neural tube from antiparallel morphogen gradients. *Science* 356: 1379-1383
- Zhang J, Tian XJ, Zhang H, Teng Y, Li R, Bai F, Elankumaran S, Xing J (2014) TGF- β -induced epithelial-to-mesenchymal transition proceeds through stepwise activation of multiple feedback loops. *Sci Signal* 7: ra91-ra91

Thank you for sending us your revised manuscript. We have now heard back from the three reviewers who were asked to evaluate your study. As you will see the reviewers are overall satisfied with the modifications made and think that the study is now suitable for publication.

Before we can formally accept your manuscript, we would ask you to address the remaining minor issues raised by the reviewers.

On a more editorial level.

REFEREE REPORTS

Reviewer #1:

Li and colleagues use a combination of computational modelling and in vivo and in vitro experimental approaches to understand the miRNA-mediated segregation of Hoxa5 and Hoxc8-expressing motor neurons during spinal cord development. In response to the referees' comments, the authors have made considerable revisions to their manuscript that further improved its overall quality. They have also made several changes to the text that made the complex mechanism they propose easier to understand and the text more concise in my opinion. As far as I can judge, they

have also done a good job of addressing the comments of the other referees.

The authors have addressed most of my comments, e.g. by adding new data that shows that the presence of multiple miR-binding sites helps the efficient degradation of *Hoxa5* and *Hoxc8*. This was at least for miR-27 and *Hoxa5* expected given their previous work (Li et al. 2017, Nature Communications). This more extensive characterization nevertheless provides a meaningful addition to this manuscript. I still think a more direct experimental confirmation of the mechanistic details gained from their computational modelling approach about the requirement of multiple miRNA-binding sites for the emergence of bistability and hysteresis would have further improved this manuscript. This could have been done by deleting combinations of miR27 or miR196-binding sites on *Hoxa5*/*Hoxc8* in ES cells and then assessing hysteresis in response to fluctuating levels of RA in the in vitro differentiations. However, this should not distract from the fact that this is overall a thoroughly and well-executed study that proposes a novel mechanism for how miRNAs can contribute to the segregation of cell fates and the establishment of sharp boundaries during development that is consistent with several experimental observations, thereby advancing our understanding of the molecular mechanisms that orchestrate the reliable patterning of tissues during development.

Minor comments:

Figure 4A,B: Just following the logic of the provided cartoons, I think there are some mistakes with the indices of k and/or the a/b parameters. E.g. following the logic of the provided cartoon in A C should turn into r with rate akR .

Figure 4C: There is a typo in the bottom part of the grey box ("iff")

Reviewer #2:

The revised version of this manuscript is greatly improved. The authors made a serious effort to make their work more accessible. In particular, while the main text is still relatively long, the description of the different models has been streamlined and is now much easier to follow. At the same time, the shape of these models and how they are analyzed is now clearer from reading the main text alone. Importantly, the origin of the observed bistability is more clearly explained and put into the context of the relevant literature. The authors have also convincingly addressed all other points I had raised. Overall, this work is a valuable addition to the field and I am happy to support its publication.

Given the depth of this work, this may be difficult at this point but my only remaining suggestion is that, wherever possible, the authors should consider further shortening/simplifying the main text and the figures (some of which are slightly complex).

Reviewer #3:

the authors have addressed all my concerns.

The authors have made all requested editorial changes.

Thank you again for sending us your revised manuscript. We are now satisfied with the modifications made and I am pleased to inform you that your paper has been accepted for publication.

Corresponding Author Name: Tian Hong; Jun-An Chen

Manuscript Number: MSB-20-9945